# NBSP: A Neuron-Level Framework for Balancing Stability and Plasticity in Deep Reinforcement Learning

## Abstract

In contrast to the human ability to continuously acquire knowledge, agents struggle with the stability-plasticity dilemma in deep reinforcement learning (DRL), which refers to the trade-off between retaining existing skills (stability) and learning new knowledge (plasticity). Current methods focus on balancing these two aspects at the network level, lacking sufficient differentiation and fine-grained control of individual neurons. To overcome this limitation, we propose Neuron-level Balance between Stability and Plasticity (NBSP) method, by taking inspiration from the observation that specific neurons are strongly relevant to task-relevant skills. Specifically, NBSP first (1) defines and identifies RL skill neurons that are crucial for knowledge retention through a goal-oriented method, and then (2) introduces a framework by employing adaptive gradient masking and experience replay techniques targeting these neurons to preserve the encoded existing skills while enabling adaptation to new tasks. Numerous experimental results on the Meta-World and Atari benchmarks demonstrate that NBSP significantly outperforms existing approaches in balancing stability and plasticity.

## 1 Introduction

**Deep reinforcement learning (DRL)** has shown exceptional capabilities across a range of complex scenarios, such as gaming (Mnih et al., 2013), robotic manipulation (Andrychowicz et al., 2020), and autonomous driving (Kiran et al., 2021). However, most RL research focuses on agents that learn to solve individual problems rather than learn a sequence of tasks continually. Ideally, the agent must maintain its performance on previously learned tasks, referred to as **stability** (McCloskey & Cohen, 1989), while simultaneously adapting to new tasks, known as **plasticity** (Carpenter & Grossberg, 1987). However, it has been revealed that emphasizing stability may hinder the ability of agents to learn new knowledge (Nikishin et al., 2022a; Abbas et al., 2023), whereas excessive plasticity can lead to catastrophic forgetting of previously acquired knowledge (Goodfellow et al., 2015; Atkinson et al., 2021b), a challenge known as the **stability-plasticity dilemma** (eMermillod et al., 2013), which remains a fundamental and under-explored problem and is the main focus of our work.

Existing methods to strike a balance between stability and plasticity generally fall into three categories, i.e. (1) **regularization-based methods** (Kirkpatrick et al., 2017; Kumar et al., 2023), which apply penalties to parameter changes to mitigate forgetting while acquiring new knowledge; (2) **replay-based methods** (Ahn et al., 2024), which leverage past experiences to consolidate knowledge; and (3) **modularity-based methods** (Kim et al., 2023; Anand & Precup, 2024), which seek to decouple stability and plasticity or isolate different components for different tasks. Despite their contributions, these methods suffer from three key limitations: (1) They primarily operate at the network level, yet their ultimate impact manifests at the level of individual neurons. However, these methods fail to

differentiate and fine-grained control neurons based on their specific roles. Therefore, identifying and effectively utilizing task-relevant neurons remains both critical and under-explored. (2) These studies are primarily conducted within the paradigm of continual learning, thus overlooking the unique characteristics intrinsic to DRL. (3) These approaches could sometimes unnecessarily inflate model parameters, thereby introducing unwarranted complexity (Bai et al., 2023).

By analyzing the activations of neurons in the DRL network, we observe that the activations of certain neurons are strongly correlated with the task goal. For instance, Figure 1 illustrates the activation distribution of a specific neuron in the network following training on the drawer-open task from the Meta-World benchmarkYu et al. (2020). Activation of the neuron serves as a reliable predictor of whether the task is successful. Higher activation levels correspond to an increased likelihood of completing the task successfully, indicating that this neuron encodes a critical skill essential for the task. Consequently, it plays a pivotal role in retaining task-specific memory.

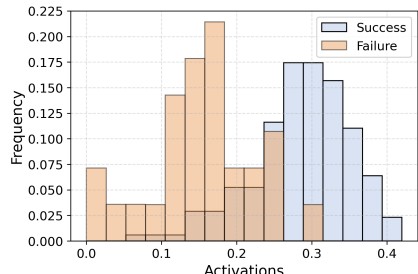

Figure 1: Distribution histogram of the activation of a neuron, categorized based on whether the drawer-open task was successfully completed or not.

Motivated by the aforementioned observations, we propose **Neuron-level Balance between Stability and Plasticity (NBSP)**, a novel DRL framework that operates at the level of neurons to tackle the stability-plasticity dilemma. In particular, (1) we first introduce **RL skill neurons**, which encode critical skills necessary for knowledge retention. While skill neurons have been investigated and successfully exploited in various domains, such as pre-trained language models (Wang et al., 2022) and neural machine translation (Bau et al., 2018), skill neurons are still much less explored in DRL. We bridge this research gap by proposing a goal-oriented strategy for identifying RL skill neurons. (2) We then apply **gradient masking** according to the scores of these neurons, ensuring that the encoded knowledge of prior skills is preserved while allowing fine-tuning during subsequent training. Meanwhile, the other neurons retain the ability to learn new tasks. (3) Additionally, we incorporate **experience replay** to periodically revisit the past experience to reinforce stability, preventing excessive drift from previous knowledge. Integrally, NBSP offers three key advantages compared with previous methods: (1) The neuron-level processing enables finer control and greater flexibility, addressing the stability-plasticity trade-off at the most fundamental level of the network. (2) The goal-oriented approach for identifying RL skill neurons is specifically tailored to DRL. (3) This framework is simple, avoiding complex network designs or additional parameters.

We conduct experiments on the **Meta-World** (Yu et al., 2020) and **Atari** (Mnih et al., 2013) benchmarks to evaluate the effectiveness of NBSP. Our results show that NBSP outperforms existing methods in balancing stability and plasticity, enabling effective learning of new tasks while preserving knowledge from previous tasks. Additionally, we perform extensive ablation studies to investigate the contribution of different components within NBSP. Specially, we analyze the DRL agents by dissecting the performance of the two critical modules, i.e., the actor and the critic. Our findings reveal that (1) addressing both the actor and critic networks yields the best performance, and (2) the critic plays a more critical role in achieving this balance due to the differences in their inherent training mechanisms. In summary, our key contributions include:

- We are the first to introduce the concept of RL skill neurons which encode skills of the task, essential for knowledge retention, and propose a goal-oriented strategy specifically tailored to DRL for identification.
- We tackle the stability-plasticity dilemma in DRL from the perspective of RL skill neurons, by employing gradient masking and experience replay on these neurons, eliminating requirements of complex network designs or additional parameters.
- We conduct extensive experiments on the Meta-World and Atari benchmarks to demonstrate the effectiveness of our method in balancing stability and plasticity.

## 2    Related Work

**Balance between stability and plasticity**. In DRL, addressing the stability-plasticity dilemma (Carpenter & Grossberg, 1988) has inspired various strategies. Stability-focused methods often utilize

replay techniques, such as A-GEM (Chaudhry et al., 2018b), using episodic memory to constrain loss, and ClonEx-SAC (Wolczyk et al., 2022), enhancing performance through behavior cloning. Pseudo-rehearsals from generative models further reduce storage requirements (Atkinson et al., 2021a). Plasticity-focused methods aim to preserve network expressiveness, with solutions like CBP (Dohare et al., 2024), resetting (Nikishin et al., 2022b), plasticity injection (Nikishin et al., 2024), Reset & Distillation (Ahn et al., 2024), and CRelu (Abbas et al., 2023) to prevent activation collapse. Modularity-based methods balance stability and plasticity by decoupling task-specific knowledge, exemplified by soft modularity for routing networks (Yang et al., 2020), value function decomposition (Anand & Precup, 2024), and compositional frameworks leveraging neural components (Mendez et al., 2022). Methods such as CRelu and ClonEx-SAC focus on continual reinforcement learning(CRL), but our study specifically targets the intrinsic balance between stability and plasticity, with other factors such as task order controlled in a cycling task setting. Moreover, while most methods operate at the network level, our approach explores neuron-level research, providing fine-grained control.

**Neuron-level research**. Recent research has shown that neuron sparsity often correlates with task-specific performance (Xu et al., 2024), driving a growing focus on skill neurons to interpret network behavior and tackle challenges across domains. For example, skill neurons have been used to enhance transferability and efficiency in Transformers via pruning (Wang et al., 2022), and dormant neurons have been recycled to improve training in DRL(Sokar et al., 2023). Other studies, such as identifying Rosetta Neurons (Dravid et al., 2023) and language-specific neurons (Tang et al., 2024), have advanced alignment and interpretability. However, neuron-level studies in DRL are still limited, with methods like CoTASP (Yang et al., 2023) and PackNet (Mallya & Lazebnik, 2018) focusing on task-specific sub-network selection via sparse prompts, pruning, and re-training. And NPC (Paik et al., 2019) restricts important neurons to maintain stability. In contrast, our work identifies RL skill neurons specific to DRL, balancing stability and plasticity with encoded task-relevant knowledge.

# 3 Methodology

In this section, we first introduce the terminology of RL skill neurons and then propose the Neuron-level Balance between Stability and Plasticity (NBSP) method.

## 3.1 Problem Setup

We focus on the setting of sequential task learning without constraints on the time intervals between tasks. In this setting, the agent is expected to perform all previously learned tasks after training, without relying on task-specific signals. For instance, large models such as DeepSeek employ RL to enhance their reasoning capabilities. However, different tasks, such as vision and mathematics, demand distinct reasoning abilities. To first strengthen a specific type of reasoning and then generalize to others, it is essential to strike a balance between stability and plasticity during sequential training. Furthermore, in real-world applications, the enhanced model should be able to handle all tasks without relying on explicit task signals. Let $\tau \in \{\tau_1, \tau_2, ...\}$ represent a sequence of task, each task $\tau$ corresponds to a distinct Markov Decision Process (MDP) $M^\tau = (S^\tau, A^\tau, P^\tau, R^\tau, \gamma^\tau)$, where $S^\tau$, $A^\tau$, $P^\tau$, $R^\tau$ and $\gamma^\tau$ denote the state space, action space, transition dynamics, reward function, and discount factor, respectively. Instead of addressing a single MDP, the goal is to solve a sequence of MDPs one by one using a universal policy $\pi(a|s)$ and Q-function $Q(s, a)$. The primary challenge lies in balancing plasticity, which refers to maximizing the discounted return of the current task, and stability, which emphasizes the maximization of the expected discounted return averaged across all previous tasks. This trade-off constitutes the core problem addressed in this work.

## 3.2 Identifying RL Skill Neurons

In this study, we make a key observation that the stability and plasticity of the agent network are closely related to its expressive capabilities, which are significantly influenced by the behavior of individual neurons. As evidenced in Molchanov et al. (2022), neuron expression determines how information is propagated and processed, directly affecting the learning and knowledge retention capabilities of the network. Therefore, understanding and controlling neuron behavior is at the most fundamental level for striking a balance between stability and plasticity. On the one hand, when neuron expression is stable and generalized, the agent network tends to exhibit high stability. On the other hand, strong plasticity can be achieved given neuron expression is flexible and adaptable.

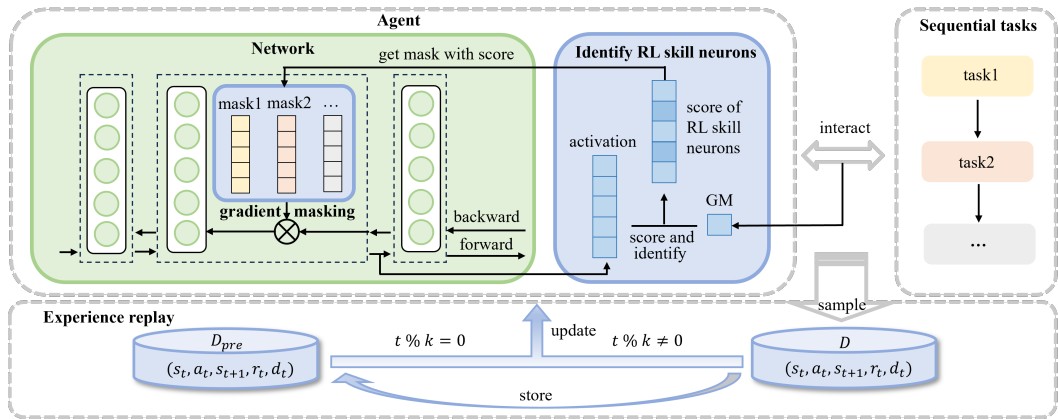

Figure 2: Framework of NBSP. The agent scores and identifies RL skill neurons for each task by measuring the activation in relation to the GM. While learning new tasks, the gradient of these neurons is masked adaptively based on their scores to preserve the encoded skills, while still allowing fine-tuning for new task learning. Additionally, a replay buffer is used to store a portion of the experiences from previous tasks, which is periodically sampled to update the agent.

Several works have demonstrated the multifaceted capabilities of neurons, such as the storage of factual knowledge (Dai et al., 2022), the association with specific languages (Tang et al., 2024), and the encoding of safety information (Chen et al., 2024). These specialized neurons, often referred as skill neurons, have been shown to significantly contribute to network performance (Wang et al., 2022). However, the potential of skill neurons in DRL remains largely under-explored. As illustrated in Figure 1, activations of the specific neuron are strongly correlated with task success: higher activation levels increase the likelihood of successful task completion, whereas lower levels are associated with failure. ***This indicates that the activations of these neurons significantly affect agent performance, effectively encoding the critical skills required for the task. By preserving the activations of such neurons, it becomes possible to retain the learned task-specific skills, thereby improving stability.***

In this work, we formally define these special neurons as **RL skill neurons**, which encode critical skills, essential for knowledge retention in DRL. Furthermore, we propose a goal-oriented method for the identification of these neurons. Unlike prior approaches that primarily focus on the inputs triggering neuron activations (Bau et al., 2020; Gurnee & Tegmark, 2023), our method emphasizes their impact on achieving ultimate goals, i.e. succeeding in finishing Meta-World tasks and attaining high scores in Atari games, by comparing the activation patterns of the neurons that exhibit varying performance levels. In Section 4.2, we empirically show the advantage of our goal-oriented method.

For a specific neuron $\mathcal{N}$, let $a(\mathcal{N}, t)$ represent its activation at step $t$. In fully connected layers, each output dimension corresponds to the activation of a specific neuron, whereas in convolution layers, the average of each output channel represents the activation of a neuron. To quantify activation level of a neuron $\mathcal{N}$, we define the **average activation** as:

$$\overline{a}(\mathcal{N}) = \frac{1}{T_{avg}} \sum_{t=1}^{T} a(\mathcal{N}, t), \tag{1}$$

where $T_{avg}$ represents the average step. The activation level of the neuron can then be assessed by comparing its current activation with the corresponding average activation.

To assess the performance of the agent at step $t$, we introduce the **Goal Metric (GM)**, denoted as $q(t)$. It serves as an evaluation metric for assessing the performance of the agent's network, varying based on the objective of the task. It is computed in an online manner during training. For instance, on the Meta-World benchmark, the GM is typically binary, determined by whether the episode is successful, which is computed at the end of each episode. In contrast, the GM is determined by the cumulative return of the episode for the Atari benchmark. Additionally, we define the **average Goal Metric** (GM) of the agent as follows, which serves as a baseline for evaluating the performance by comparing it with the current GM.

$$\overline{q} = \frac{1}{T_{avg}} \sum_{t=1}^{T} q(t). \tag{2}$$

To differentiate the roles of neurons across various tasks, it is essential to assess neuron activations in relation to specific goals. Intuitively, we can consider a neuron $\mathcal{N}$ to be positively contributing to the goal at step $t$ when its activation $a(\mathcal{N}, t)$ surpasses the average activation $\overline{a}(\mathcal{N})$, i.e. $a(\mathcal{N}, t) > \overline{a}(\mathcal{N})$, while the GM at the same step also exceeds its average, i.e. $q(t) > \overline{q}$. To quantify this contribution, we accumulate a batch of results over $T$ steps and define the over-activation rate as follows:

$$R_{over}(\mathcal{N}) = \frac{\sum_{t=1}^{T} 1_{[1_{[a(\mathcal{N},t)>\overline{a}(\mathcal{N})]}=1_{[q(t)>\overline{q}]}]}}{T}. \tag{3}$$

Here, $1_{[condition]} \in \{0, 1\}$ denotes the indicator function, which returns 1 if and only if the specified condition is satisfied. While Eq. (3) assesses the positive correlation of neurons towards achieving the goal, where a higher rate implies a greater significance of the neuron in producing better outcome, however, it overlooks neurons that exhibit a negative correlation with the goal but still carry valuable task-related knowledge. Specifically, when the activation of a neuron falls below its average activation, the agent performs well conversely. To this end, we define a **comprehensive score Score**$(\mathcal{N})$ for the neuron that takes into account both positive and negative effects:

$$Score(\mathcal{N}) = max(R_{over}(\mathcal{N}), 1 - R_{over}(\mathcal{N})). \tag{4}$$

Subsequently, we rank all neurons in the agent network, excluding those in the last layer, in descending order based on their scores. The RL skill neurons are determined by selecting the neurons with the top m% highest scores, formally defined as follows, where $\tau_m(\cdot)$ denotes the top-m selection operator. And the pseudo-code of the identification method is shown in Appendix D.

$$\mathcal{N}_{RL\ skill} = \tau_m(Score(\mathcal{N})) \tag{5}$$

### 3.3 Neuron-level Balance between Stability and Plasticity

Building upon the concept of RL skill neurons, we propose a novel DRL framework — **Neuron-level Balance between Stability and Plasticity (NBSP)**, as shown in Figure 2. Unlike prior methods (Bai et al., 2023; Kim et al., 2023), the framework proposed does not require complex network designs or additional parameters. Given that RL skill neurons encode essential task-specific skills, preserving their activation patterns is critical to maintaining knowledge from previous tasks during continual tasks learning. However, simply freezing RL skill neurons would hinder the ability of the agent to adapt to new tasks. To address this challenge, NBSP employs an adaptive **gradient masking** technique. Specifically, during each update round in the continual learning process, the gradients of RL skill neurons are selectively masked to restrict changes in their activation patterns while allowing other neurons to adapt freely. This process is formally expressed as follows:

$$\Delta W_{:,j} = mask_j^{(l)} \cdot \Delta W_{:,j}^{(l)}, \tag{6}$$

where $\Delta W_{:,j}^{(l)}$ denotes the gradient with respect to the weight $W_{:,j}^{(l)}$ in the $l$-th layer of the network, and $j$ is the index of the output neuron in that layer. The term $mask_j^{(l)}$ is associated with the score of $j$-th neuron in the $l$-th layer, which could be calculated as follows:

$$mask(\mathcal{N}) = \begin{cases} \alpha(1 - Score(\mathcal{N})) & \text{if } \mathcal{N} \in \mathcal{N}_{RL\ skill} \\ 1 & \text{if } \mathcal{N} \notin \mathcal{N}_{RL\ skill} \end{cases}, \tag{7}$$

where $\mathcal{N}_{RL\ skill}$ represents the set of RL skill neurons, and $\alpha$ is a super-parameter that determines the degree of restriction on these neurons, which is configured to 0.2 in the experiment. ***By employing gradient masking, NBSP effectively safeguards the encoded skills within RL skill neurons from interference during the learning of new tasks, thereby enhancing stability. At the same time, RL skill neurons remain adaptable, allowing fine-tuning to accommodate new tasks and maintaining high plasticity. In addition, neurons except RL skill neurons are free to fully engage in learning new task-specific knowledge, ensuring comprehensive learning across tasks.***

To mitigate excessive drift from knowledge acquired in previous tasks, we integrate the **experience replay** technique, periodically sampling prior experiences at specific intervals $k$. After training on a task, a portion of the experiences, rather than the entirety, are stored in a unified replay buffer $D_{pre}$, requiring only a modest memory footprint. By incorporating experience replay, the stability of DRL agents is further enhanced. The corresponding loss function is defined as follows:

$$\mathcal{L} = R(t) \cdot \mathbb{E}_{(s_t,a_t,s_{t+1},r_t,d_t) \sim D_{pre}}[L] + (1 - R(t)) \cdot \mathbb{E}_{(s_t,a_t,s_{t+1},r_t,d_t) \sim D}[L], \tag{8}$$

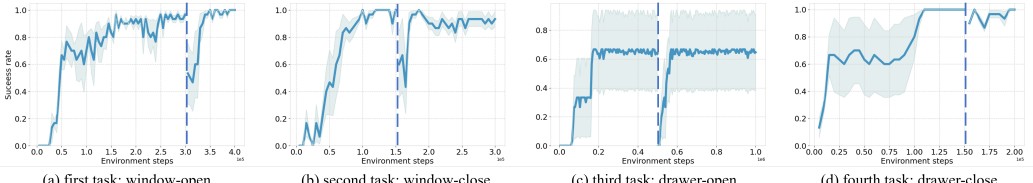

| (a) first task: window-open | (b) second task: window-close | (c) third task: drawer-open | (d) fourth task: drawer-close |

Figure 3: Training process of NBSP on the Meta-World benchmark. The segments to the left and right of the dashed line represent the training processes of the first and second cycles, respectively.

where $L$ denotes the original loss function, $R(t)$ is a binary function that evaluates to 1 if and only if the current step $t$ is at an interval. $D$ represents the replay buffer for the current task, and $(s_t, a_t, s_{t+1}, r_t, d_t)$ denotes the tuple of the current state, action, next state, reward, and whether the episode is done sampled from the replay buffer. The pseudo-code of NBSP is shown in Appendix D.

## 4 Experiment

In this section, we evaluate the performance of NBSP on the **Meta-World** (Yu et al., 2020) and **Atari** benchmarks (Mnih et al., 2013).

Table 1: Results of NBSP with other baselines on the Meta-World benchmark.

| Cycling sequential tasks | Metrics | Methods | | | | | | | |
|---|---|---|---|---|---|---|---|---|---|
| | | EWC | NPC | ANCL | CoTASP | CRelu | CBP | PI | NBSP |
| (window-open → window-close) | ASR ↑ | $0.63 \pm 0.03$ | $0.26 \pm 0.01$ | $0.66 \pm 0.04$ | $0.05 \pm 0.01$ | $0.26 \pm 0.14$ | $0.67 \pm 0.05$ | $0.61 \pm 0.02$ | $\mathbf{0.90 \pm 0.04}$ |
| | FM ↓ | $0.89 \pm 0.07$ | $0.68 \pm 0.04$ | $0.84 \pm 0.10$ | $\mathbf{0.01 \pm 0.01}$ | $0.66 \pm 0.42$ | $0.78 \pm 0.13$ | $0.91 \pm 0.07$ | $\mathbf{0.18 \pm 0.01}$ |
| | FWT ↑ | $\mathbf{0.97 \pm 0.02}$ | $0.26 \pm 0.01$ | $0.97 \pm 0.03$ | $0.04 \pm 0.01$ | $0.33 \pm 0.19$ | $0.95 \pm 0.02$ | $0.95 \pm 0.01$ | $\mathbf{0.96 \pm 0.02}$ |
| (drawer-open → drawer-close) | ASR ↑ | $0.68 \pm 0.06$ | $0.35 \pm 0.05$ | $0.64 \pm 0.02$ | $0.07 \pm 0.01$ | $0.29 \pm 0.20$ | $0.61 \pm 0.03$ | $0.60 \pm 0.07$ | $\mathbf{0.96 \pm 0.02}$ |
| | FM ↓ | $0.80 \pm 0.15$ | $0.69 \pm 0.05$ | $0.88 \pm 0.09$ | $\mathbf{0.01 \pm 0.01}$ | $0.31 \pm 0.32$ | $0.91 \pm 0.03$ | $0.71 \pm 0.30$ | $\mathbf{0.07 \pm 0.06}$ |
| | FWT ↑ | $\mathbf{0.98 \pm 0.01}$ | $0.39 \pm 0.09$ | $0.96 \pm 0.01$ | $0.09 \pm 0.00$ | $0.42 \pm 0.28$ | $0.93 \pm 0.04$ | $0.88 \pm 0.15$ | $\mathbf{0.98 \pm 0.01}$ |
| (button-press-topdown → window-open) | ASR ↑ | $0.66 \pm 0.06$ | $0.25 \pm 0.00$ | $0.61 \pm 0.01$ | $0.03 \pm 0.00$ | $0.33 \pm 0.10$ | $0.62 \pm 0.01$ | $0.63 \pm 0.02$ | $\mathbf{0.95 \pm 0.05}$ |
| | FM ↓ | $0.85 \pm 0.14$ | $0.67 \pm 0.00$ | $0.95 \pm 0.05$ | $\mathbf{0.01 \pm 0.00}$ | $0.94 \pm 0.01$ | $0.97 \pm 0.03$ | $0.97 \pm 0.05$ | $\mathbf{0.08 \pm 0.12}$ |
| | FWT ↑ | $0.96 \pm 0.01$ | $0.25 \pm 0.01$ | $0.95 \pm 0.03$ | $0.04 \pm 0.01$ | $0.42 \pm 0.20$ | $\mathbf{0.98 \pm 0.02}$ | $\mathbf{0.98 \pm 0.02}$ | $\mathbf{0.98 \pm 0.01}$ |
| (window-open → window-close → drawer-open → drawer-close) | ASR ↑ | $0.44 \pm 0.05$ | $0.19 \pm 0.04$ | $0.48 \pm 0.04$ | $0.04 \pm 0.01$ | $0.10 \pm 0.06$ | $0.43 \pm 0.03$ | $0.41 \pm 0.06$ | $\mathbf{0.66 \pm 0.14}$ |
| | FM ↓ | $0.74 \pm 0.11$ | $0.50 \pm 0.02$ | $0.80 \pm 0.04$ | $\mathbf{0.04 \pm 0.01}$ | $0.39 \pm 0.02$ | $0.91 \pm 0.05$ | $0.84 \pm 0.05$ | $\mathbf{0.48 \pm 0.18}$ |
| | FWT ↑ | $0.83 \pm 0.10$ | $0.20 \pm 0.05$ | $0.89 \pm 0.06$ | $0.08 \pm 0.01$ | $0.13 \pm 0.10$ | $\mathbf{0.97 \pm 0.02}$ | $0.82 \pm 0.10$ | $\mathbf{0.89 \pm 0.12}$ |
| (button-press-topdown → window-close → door-open → drawer-close) | ASR ↑ | $0.43 \pm 0.03$ | $0.17 \pm 0.01$ | $0.44 \pm 0.03$ | $0.04 \pm 0.01$ | $0.14 \pm 0.11$ | $0.41 \pm 0.02$ | $0.38 \pm 0.01$ | $\mathbf{0.74 \pm 0.07}$ |
| | FM ↓ | $0.81 \pm 0.09$ | $0.47 \pm 0.01$ | $0.87 \pm 0.02$ | $\mathbf{0.04 \pm 0.00}$ | $0.62 \pm 0.16$ | $0.94 \pm 0.02$ | $0.97 \pm 0.02$ | $\mathbf{0.34 \pm 0.15}$ |
| | FWT ↑ | $0.88 \pm 0.10$ | $0.19 \pm 0.02$ | $0.91 \pm 0.08$ | $0.07 \pm 0.02$ | $0.17 \pm 0.15$ | $\mathbf{0.97 \pm 0.01}$ | $0.92 \pm 0.07$ | $\mathbf{0.95 \pm 0.06}$ |

**Experiment setting**. We follow the the experimental paradigm of Abbas et al. (2023); Liu et al. (2024), evaluating our proposed method on a **cycling sequence of tasks** characterized by non-stationarity due to changing environments over time. Specifically, the agent learns each task sequentially and transitions to the next without resetting the learned networks. The task cycles through a fixed sequence, with a cycle completing once all tasks in the sequence have been learned. The agent cycles twice, resulting in each task being repeated twice during the training process. Compared to the CRL training paradigm, our cycling training paradigm provides a more specific evaluation of the balance between stability and plasticity. By repeating each task twice within a cycling sequence, the setup not only assesses the plasticity in adapting to new tasks but also evaluates its stability when revisiting previously learned tasks, avoiding the influence of task order. Details about the benchmarks are shown in Appendix C.2.

For all experiments, we use the Soft Actor-Critic (SAC) (Haarnoja et al., 2018) algorithm, as implemented by CleanRL (Huang et al., 2022). Each agent is trained until either reaching a predefined maximum number of steps or demonstrating stable mastery of the task in the Meta-World benchmark. Each experiment is repeated using three different random seeds. The shaded regions in the figures and the plus/minus numbers represent the standard error across multiple seeds. Detailed descriptions of the hyper-parameters and other experimental settings are provided in Appendix C.3.

**Metric**. Overall performance is commonly assessed using the **Average Success Rate (ASR)**, analogous to the AIA metric (Wang et al., 2024). Let $sr_{i,j}$ represent the success rate on the $j$-th task

after completing the learning of the $i$-th task ($i \geq j$), $H$ denote the number of tasks. The ASR is
defined as follows. The higher the ASR, the better the method balances stability and plasticity.

$$ASR = \frac{1}{H} \sum_{i=1}^{H} \frac{1}{i} \sum_{i \geq j} sr_{i,j}, \tag{9}$$

To evaluate the stability of the agent, we utilize the **Forgetting Measure (FM)** (Chaudhry et al.,
2018a). The lower the FM, the better the method maintains stability, which is calculated as:

$$FM = \frac{1}{H-1} \sum_{i=2}^{H} \frac{1}{i-1} \sum_{i \geq j} \underset{l \in \{1,...,i-1\}}{max} (sr_{l,j} - sr_{i,j}). \tag{10}$$

To assess the plasticity of the agent, we employ the **Forward Transfer (FWT)** metric (Lopez-Paz &
Ranzato, 2017), which is calculated as follows:

$$FWT = \frac{1}{H} \sum_{i=1}^{H} sr_{i,i}. \tag{11}$$

The higher the FWT, the better the method maintains plasticity. Further details about evaluation
metrics are available in Appendix C.4.

**Baseline**. To assess the effectiveness of our proposed NBSP framework, we compare it with seven
baseline methods dealing with the balance between stability and plasticity. **EWC** (Kirkpatrick et al.,
2017) and **NPC** (Paik et al., 2019) primarily emphasize maintaining stability, while **CRelu** (Abbas
et al., 2023), **CBP** (Dohare et al., 2024), and **PI** (Nikishin et al., 2024) focus on enhancing plasticity.
**ANCL** (Kim et al., 2023) and **CoTASP** (Yang et al., 2023) aim to achieve a balance between stability
and plasticity. Notably, CoTASP makes relevant tasks share more neurons in the meta-policy network,
and NPC estimates the importance value of each neuron and consolidates important neurons, they are
both relevant to neurons. Detailed descriptions of these baselines can be found in Appendix C.1.

### 4.1 Experiment on the Meta-World Benchmark

The experimental results of NBSP compared with other baselines on the Meta-World benchmark
are presented in Table 1. As shown in the final column, NBSP significantly outperforms all other
methods in the overall performance metric ASR. For two-task cycling tasks, NBSP achieves an ASR
consistently above 0.9, which is substantially higher than other baselines. Its stability metric, FM, is
markedly lower, while its plasticity metric, FWT, remains at a high level. Furthermore, NBSP also
demonstrates excellent performance in four-task cycling tasks, maintaining a substantial lead.

For stability-focused baselines, EWC achieves a relatively good ASR compared to other baselines
but still falls short of NBSP. Moreover, EWC exhibits poor stability due to its high FM values. NPC
performs even worse, failing to maintain both stability and plasticity effectively. Among plasticity-
focused baselines, CBP and PI achieve comparable plasticity to NBSP, as reflected in their high FWT
scores. However, both suffer from severe stability loss, indicated by their higher FM values. Another
plasticity-focused method, CRelu, underperforms in both stability and plasticity. For baselines
attempting to balance stability and plasticity, ANCL achieves high plasticity with competitive FWT
scores but fails to retain prior knowledge, as reflected by its high FM value. CoTASP, despite being
explicitly designed for this trade-off, performs poorly overall. Its low FM is attributed to a failure to
acquire meaningful task knowledge, as evidenced by its low FWT value.

The effectiveness of NBSP is further demonstrated in Figure 3, which showcases the training dynamics
of NBSP. Specifically, during the second cycle of learning the same task, the agent exhibits a high
success rate even before retraining, indicating that it has retained significant task knowledge. As
a result, the agent is able to master the task more rapidly. This highlights the ability of NBSP to
preserve knowledge from prior tasks while simultaneously maintaining the plasticity required to learn
new tasks effectively. The other training process is demonstrated in Appendix C.7. In summary,
***NBSP delivers a remarkable improvement in maintaining stability without compromising plasticity,***
***achieving a well-balanced trade-off in DRL.***

### 4.2 Ablation Study

In the ablation study, we further evaluate the effectiveness of (1) the two primary components of
NBSP: the gradient masking technique and experience replay technique, (2) the neuron identification

Table 2: Results of ablation study of gradient masking and experience replay techniques.

| Metrics | (button-press-topdown → window-open) | | | | |
|---|---|---|---|---|---|
| | vanilla SAC | only experience replay | only gradient masking | NBSP with hard gradient masking | NBSP |
| ASR ↑ | 0.62 ± 0.01 | 0.70 ± 0.08 | 0.71 ± 0.06 | 0.71±0.03 | **0.95 ± 0.05** |
| FM ↓ | 0.99 ± 0.02 | 0.50 ± 0.16 | 0.73 ± 0.21 | 0.72±0.04 | **0.08 ± 0.12** |
| FWT ↑ | **0.98 ± 0.02** | 0.92 ± 0.05 | 0.97 ± 0.02 | **0.98±0.03** | **0.98 ± 0.01** |

Table 3: Results of ablation study of neuron identification methods.

| Metrics | (window-open → window-close) | | | | (drawer-open → drawer-close) | | | | (button-press-topdown → window-open) | | | |
|---|---|---|---|---|---|---|---|---|---|---|---|---|
| | activation | weight | random | ours | activation | weight | random | ours | activation | weight | random | ours |
| ASR ↑ | 0.65±0.30 | 0.73±0.20 | 0.78±0.09 | **0.90±0.04** | 0.82±0.06 | 0.51±0.17 | 0.72±0.26 | **0.96±0.02** | 0.75±0.01 | 0.93±0.06 | 0.72±0.01 | **0.95±0.05** |
| FM ↓ | 0.56±0.37 | 0.44±0.31 | 0.42±0.13 | **0.18±0.01** | 0.44±0.16 | 0.67±0.00 | 0.41±0.28 | **0.07±0.06** | 0.65±0.02 | 0.15±0.12 | 0.70±0.05 | **0.08±0.12** |
| FWT ↑ | 0.73±0.35 | 0.81±0.22 | 0.90±0.06 | **0.96±0.02** | 0.98±0.02 | 0.69±0.22 | 0.83±0.23 | **0.98±0.01** | **0.99±0.00** | 0.98±0.02 | 0.96±0.02 | 0.98±0.01 |

method, and (3) the two critical modules of DRL: the actor and the critic. What's more, we analyze how the proportion of RL skill neurons influences the performance of NBSP.

**Gradient masking and experience replay**. To evaluate the contributions of the two core components of NBSP, we designed five experimental settings: (1) vanilla SAC, (2) SAC with only the experience replay, (3) SAC with only the gradient masking, (4) SAC with experience replay and hard gradient masking, where the masks of RL skill neurons are set directly to zero, and (5) NBSP.

The results of the cycling sequential tasks (button-press-topdown → window-open) are shown in Table 2. From the results, we observe the following: (1) The vanilla SAC algorithm suffers from severe stability loss, as indicated by a high FM score, underscoring the need for mechanisms to retain prior knowledge. (2) Using either experience replay or gradient masking alone alleviates the stability loss to some extent, confirming their individual effectiveness. (3) Combining both techniques in NBSP significantly improves performance, with lower FM (indicating enhanced stability) and higher FWT (demonstrating maintained plasticity). (4) Our adaptive gradient masking, which sets masks of RL skill neurons based on their scores, outperforms hard masking (setting masks to zero directly), demonstrating its superior effectiveness. ***These findings demonstrate that neither experience replay nor gradient masking alone can properly balance stability and plasticity, while their combination achieves optimal performance.*** The reason is that gradient masking and experience replay focus on different mechanisms and therefore complement each other. Gradient masking primarily targets RL skill neurons to reduce interference with past knowledge while maintaining the ability to fine-tune for new tasks. And experience replay mainly acts on neurons except RL skill neurons to prevents these neurons from being overly biased toward new tasks. Additional results for different task settings are provided in Appendix C.8.

**Neuron identification method**. To evaluate the proposed goal-oriented neuron identification method, we compare it with three alternative strategies: (1) random neuron identification, (2) identifying neurons with activation magnitude (Jung et al., 2020), and (3) identifying neurons with weight magnitude (Dohare et al., 2021). As shown in Table 3, our goal-oriented method consistently outperforms the other three methods across all three metrics: ASR, FM, and FWT, which confirms that our method effectively identifies neurons critical for knowledge retention, ensuring better stability and plasticity in cycling sequential task learning. ***These findings validate the necessity of task-specific, goal-oriented neuron identification in enhancing balance between stability and plasticity.***

**Actor and critic**. To get a deeper understanding of the individual roles of the actor and critic in DRL agents, we compare the performance of NBSP with that only applied on actor and critic. The result is shown in Table 4. ***The results indicate that both the actor and critic networks are essential for striking an optimal balance between stability and plasticity. Notably, the critic proves to be the more critical module in balancing this trade-off***, which aligns with the insight from Ma et al. (2024) that plasticity loss in the critic serves as the principal bottleneck impeding efficient training in DRL. ***We further investigate this phenomenon by dissecting the inherent training mechanisms of actor-critic RL methods, and draw the following key observations***: (1) Updates to the actor are guided by feedback from the critic. Consequently, even if the RL skill neurons in the actor are masked, they remain influenced by the critic, which may gradually adapt to the new task at the expense of retaining prior knowledge; (2) In contrast, applying NBSP to the critic network indirectly constrains the actor as well; and (3) The update process of the critic network is recursive, with its target network updated via an exponential moving average, enabling it to preserve knowledge from the previous task while integrating new skills. Therefore, NBSP achieves better performance on the critic than on the

Table 4: Results of ablation study of the actor and critic modules.

| Metric | (window-open → window-close) | | | (drawer-open → drawer-close) | | | (button-press-topdown → window-open) | | |
|---|---|---|---|---|---|---|---|---|---|
| | actor | critic | both | actor | critic | both | actor | critic | both |
| ASR ↑ | 0.76 ± 0.10 | 0.79 ± 0.05 | **0.90 ± 0.04** | 0.79 ± 0.05 | 0.86 ± 0.02 | **0.96 ± 0.02** | 0.81 ± 0.11 | 0.85 ± 0.16 | **0.95 ± 0.05** |
| FM ↓ | 0.58 ± 0.19 | 0.48 ± 0.09 | **0.18 ± 0.01** | 0.55 ± 0.15 | 0.31 ± 0.03 | **0.07 ± 0.06** | 0.45 ± 0.28 | 0.35 ± 0.38 | **0.08 ± 0.12** |
| FWT ↑ | **0.97 ± 0.04** | 0.94 ± 0.05 | 0.96 ± 0.02 | **0.99 ± 0.01** | 0.96 ± 0.02 | 0.98 ± 0.01 | 0.95 ± 0.01 | 0.95 ± 0.03 | **0.98 ± 0.01** |

Table 5: Results of NBSP with other baselines on the Atari benchmark.

| Cycling sequential games | Metrics | Methods | | | | | | | |
|---|---|---|---|---|---|---|---|---|---|
| | | EWC | NPC | ANCL | CoTASP | CRelu | CBP | PI | NBSP |
| (Pong → Bowling) | AR ↑ | 0.66 ± 0.07 | 0.51 ± 0.02 | 0.42 ± 0.29 | -0.05 ± 0.02 | 0.02 ± 0.00 | -0.09 ± 0.00 | 0.53 ± 0.01 | **0.87 ± 0.01** |
| | FM ↓ | 0.58 ± 0.20 | 0.51 ± 0.04 | 0.46 ± 0.31 | 0.07 ± 0.01 | **0.01 ± 0.00** | 0.06 ± 0.00 | 0.78 ± 0.02 | 0.05 ± 0.03 |
| | FWT ↑ | 0.70 ± 0.02 | 0.35 ± 0.02 | 0.47 ± 0.31 | -0.05 ± 0.05 | 0.02 ± 0.01 | -0.09 ± 0.00 | 0.60 ± 0.00 | **0.72 ± 0.01** |
| (BankHeist → Alien) | AR ↑ | 0.46 ± 0.01 | 0.38 ± 0.06 | 0.46 ± 0.01 | -0.08 ± 0.05 | 0.08 ± 0.05 | 0.12 ± 0.02 | 0.48 ± 0.14 | **0.57 ± 0.02** |
| | FM ↓ | 0.98 ± 0.02 | 0.46 ± 0.14 | 0.98 ± 0.03 | **0.27 ± 0.04** | 0.52 ± 0.29 | 0.44 ± 0.09 | 0.88 ± 0.27 | 0.65 ± 0.07 |
| | FWT ↑ | 0.71 ± 0.02 | 0.37 ± 0.03 | 0.72 ± 0.01 | -0.16 ± 0.07 | 0.28 ± 0.11 | 0.30 ± 0.05 | **0.73 ± 0.26** | 0.72 ± 0.05 |

actor. This demonstrates the distinct roles of the actor and critic in balancing stability and plasticity, providing valuable insights for future research in this field.

**The proportion of RL skill neurons**. To evaluate the impact of the proportion of RL skill neurons on the performance of NBSP, we experiment with various proportions on the (button-press-topdown → window-open) cycling tasks. The results, shown in Figure 4, reveal an interesting trend: *as the proportion of RL skill neurons increases, the ASR improves initially, but begins to decline after reaching a certain threshold*. Specifically, when the proportion is small, not all neurons encoding task-specific

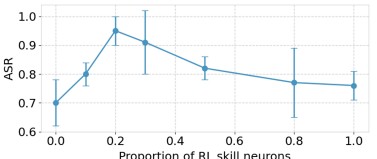

Figure 4: Performance of NBSP with different proportions of RL skill neurons.

skills are identified, leading to knowledge loss stored in neurons that are not selected. On the other hand, when the proportion becomes too large, neurons that do not encode skills may be incorrectly selected as RL skill neurons, which compromises their learning capacity and causes the true RL skill neurons to adjust their activations to accommodate new tasks, ultimately reducing stability. Thus, determining the optimal proportion of RL skill neurons is crucial for achieving the best performance. Our experiments suggest that a proportion of 0.2 is ideal for balancing stability and plasticity.

## 4.3 Experiment on the Atari Benchmark

We further evaluate NBSP on the Atari benchmark to assess its generalization ability. In contrast to the continuous action space of Meta-World, Atari games feature discrete action spaces, and episode returns are used to evaluate the performance of each game. The results are presented in Table 5. As with the Meta-World benchmark, NBSP demonstrates superior performance in balancing stability and plasticity, outperforming other baselines across key evaluation metrics, including AR (Average Return), FM, and FWT. In a word, *NBSP exhibits excellent generalization in balance stability and plasticity across different benchmarks.*

## 5 Conclusion

This work addresses the fundamental issue of the stability-plasticity dilemma in DRL. To tackle this problem, we introduce the concept of RL skill neurons by identifying neurons that significantly contribute to knowledge retention, building upon which we then propose the Neuron-level Balance between Stability and Plasticity framework, by employing gradient masking and experience replay techniques on RL skill neurons. Experimental results on the Meta-World and Atari benchmarks demonstrate that NBSP significantly outperforms existing methods in managing the stability-plasticity trade-off. Future research could explore the application of RL skill neurons like model distillation and extend NBSP to other learning paradigms, such as supervised learning.

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

## A Related Wrok

**Balance between stability and plasticity**. In DRL, the agent faces a fundamental challenge: the stability-plasticity dilemma, first introduced by Carpenter & Grossberg (1988). Recent research has proposed various strategies to address this issue by balancing stability and plasticity.

Replay-based methods are widely employed to enhance stability by reusing experiences from past distributions. For example, Chaudhry et al. (2018b) introduced A-GEM, which combines episodic memory to ensure that the average loss of prior tasks does not increase when learning a new task. Similarly, Wolczyk et al. (2022) proposed ClonEx-SAC, which uses actor behavioral cloning and best-return exploration to boost performance in CRL. To reduce storage requirements, pseudo-rehearsals generated from a generative model have also been proposed (Atkinson et al., 2021a).

Maintaining the expressiveness of neurons is key to preserving plasticity. Nikishin et al. (2022b) proposed a mechanism that periodically resets a portion of the agent's network to counteract plasticity loss. Likewise, Nikishin et al. (2024) introduced plasticity injection, a lightweight intervention that enhances network plasticity without increasing trainable parameters or introducing prediction bias. The Reset & Distillation (R&D) framework combines resetting the online actor-critic network for new tasks with offline distillation of knowledge from previous action probabilities, effectively retaining plasticity (Ahn et al., 2024). Additionally, Abbas et al. (2023) proposed the Concatenated ReLUs (CReLUs) activation function to prevent activation collapse, thereby alleviating plasticity degradation.

Modularity-based approaches have shown promise in balancing stability and plasticity by decoupling task-specific and general knowledge. For instance, Anand & Precup (2024) decomposed the value function into a permanent value function, which captures persistent knowledge, and a transient value function, which facilitates rapid adaptation. Yang et al. (2020) designed a routing network to estimate task-specific routing strategies, reconfigure the base network, and combine routes using a soft modularity mechanism, making it effective for sequential tasks. Similarly, Mendez et al. (2022) proposed a compositional lifelong RL framework that uses accumulated neural components to accelerate learning for new tasks while preserving performance on past tasks via offline RL and replayed experiences.

**Neuron-level Research** Recent research highlights that not all neurons remain active across varying contexts, and this neuron sparsity is often positively correlated with task-specific performance (Xu et al., 2024). Building on this insight, numerous studies have focused on identifying and leveraging skill neurons to interpret network behavior and tackle specific challenges, achieving significant advancements. For example, skill neurons in pre-trained Transformers, which demonstrate strong predictive value for task labels, have been utilized for network pruning to enhance efficiency and improve transferability (Wang et al., 2022). Sokar et al. (2023) investigate dormant neurons in deep reinforcement learning and propose a method to recycle them during training. Similarly, Dravid et al. (2023) introduce Rosetta Neurons, enabling cross-class alignments and transformations without specialized training. In large language models, language-specific neurons have been identified to control output languages by selective activation or deactivation (Tang et al., 2024), while safety neurons have been analyzed to enhance safety alignment through mechanistic interpretability (Chen et al., 2024).

Despite these achievements, the exploration of skill neurons in DRL remains limited. Existing neuron-level approaches primarily focus on task-specific sub-network selection. For instance, CoTASP learns hierarchical dictionaries and meta-policies to generate sparse prompts and extract sub-networks as task-specific policies (Yang et al., 2023). Similarly, Mallya & Lazebnik (2018) sequentially allocate multiple tasks within a single network through iterative pruning and re-training, balancing performance and storage efficiency. Unlike these methods, our work identifies RL skill neurons specifically tailored to deep reinforcement learning, ensuring a balance between stability and plasticity by preserving the task-relevant knowledge encoded in these neurons while allowing for fine-tuning.

## B Preliminary

### B.1 Markov Decision Process (MDP)

A Markov Decision Process(MDP) is a framework used to describe a problem involving learning from actions to achieve a goal. Almost all reinforcement learning problems can be characterized

as a Markov Decision Process. Each MDP is defined by a tuple $< S, A, P, R, \gamma >$, where $S$ and $A$ represent state and action spaces respectively. The transition dynamics of the MDP are defined by the function $P : S \times A \times S \to [0, 1]$, which represents the probability of transitioning from a give state $s$ with action $a$ to state $s'$. The reward function is represented by $R : S \times A \times S \to \mathbb{R}$, and $\gamma \in (0, 1)$ is the discount factor. At each time step $t$, an agent observes the state of the environment, denoted as $s_t$, and selects an action $a_t$ according to a policy $\pi(a|s)$. One time step later, the agent receives a numerical reward $r_{t+1}$ and transitions to a new state $s_{t+1}$. In the simplest case, the return is the sum of the rewards when the agent–environment interaction naturally breaks into subsequences, which we refer to episodes (Sutton, 2018).

## B.2 Soft Actor-Critic (SAC)

Soft Actor-Critic (SAC) is an off-policy actor-critic deep reinforcement learning algorithm that leverages maximum entropy to promote exploration. This work employs SAC to train a policy that effectively balances stability and plasticity , chosen for its sample efficiency, excellent performance, and robust stability. In this framework, the actor aims to maximize both the expected reward and the entropy of the policy. The parameters $\phi$ of the actor are optimized by minimizing the following loss function:

$$J_\pi(\phi) = E_{s_t \sim D, a_t \sim \pi_\phi}[\alpha log \pi_\phi(a_t|s_t) - Q_\theta(s_t, a_t)],$$

where $D$ is the replay buffer, $\alpha$ is the temperature parameter controlling the trade-off between exploration and exploitation, $\theta$ denotes the parameters of the critic network, $\pi_\phi$ represents the policy learned by the actor $\phi$ , and $Q_\theta$ denotes the Q-value estimated by the critic $\theta$. The critic network is trained to minimize the squared residual error:

$$J_Q(\theta) = E_{(s_t, a_t, s_{t+1}) \sim D}[\frac{1}{2}(Q_\theta(s_t, a_t) - r_t - \gamma \hat{V}(s_{t+1})],$$

$$\hat{V}(s_t) = E_{a_t \sim \pi_\phi}[Q_\theta(s_t, a_t) - \alpha log \pi_\phi(a_t|s_t)],$$

where $\gamma$ represents the discount factor.

## B.3 Neuron

In neural networks, various components, such as blocks and layers, play distinct roles. Here, we define a neuron as a single output dimension from a layer. For example, in a fully connected layer, each output dimension corresponds to a neuron. Similarly, in a convolutional layer, each output channel represents a neuron. Furthermore, following the terminology used by Sajjad et al. (2022), we classify neurons that encapsulate a single concept as focused neurons, while a group of neurons collectively representing a concept are termed group neurons.

# C  Experiment

## C.1  Baseline

**EWC**: Elastic Weight Consolidation (EWC) (Kirkpatrick et al., 2017) addresses the challenge of catastrophic forgetting by allowing neural networks to retain proficiency in previously learned tasks even after a long hiatus. It achieves this by selectively slowing down learning for weights that are crucial for retaining knowledge of these tasks. This approach has demonstrated excellent performance in sequentially solving a series of classification tasks, such as those in the MNIST handwritten digit dataset, and in learning several Atari 2600 games sequentially.

**NPC**: Neuron-level Plasticity Control (NPC) (Paik et al., 2019) preserves the existing knowledge from the previous tasks by controlling the plasticity of the network at the neuron level. NPC estimates the importance value of each neuron and consolidates important neurons by applying lower learning rates, rather than restricting individual connection weights to stay close to the values optimized for the previous tasks. The experimental results on the several classification datasets show that neuron-level consolidation is substantially effective.

**ANCL**: Auxiliary Network Continual Learning (ANCL) is an innovative approach that incorporates an auxiliary network to enhance plasticity within a model that primarily emphasizes stability. Specifically,

this framework introduces a regularizer that effectively balances plasticity and stability, achieving superior performance over strong baselines in both task-incremental and class-incremental learning scenarios.

**CoTASP**: Continual Task Allocation via Sparse Prompting (CoTASP) (Yang et al., 2023) learns over-complete dictionaries to produce sparse masks as prompts extracting a sub-network for each task from a meta-policy network. Hence, relevant tasks share more neurons in the meta-policy network due to similar prompts while cross-task interference causing forgetting is effectively restrained. It outperforms existing continual and multi-task RL methods on all seen tasks, forgetting reduction, and generalization to unseen tasks.

**CRelu**: Concatenated ReLUs (CReLUs) (Abbas et al., 2023) is a simple activation function that concatenates the input with its negation and applies ReLU to the result. It performs effectively in facilitating continual learning in a changing environment.

**CBP**: Continual BackPropagation (CBP) (Dohare et al., 2024) reinitializes a small number of units during training, typically fewer than one per step. To prevent disruption of what the network has already learned, only the least-used units are considered for reinitialization. It shows great performance on Continual ImageNet and class-incremental CIFAR-100.

**PI**: Plasticity Injection (PI) (Nikishin et al., 2024) freeze the parameters $\theta$ and introduce a new set of parameters $\theta\prime$ sampled from random initialization at some point in training, where the network might have started losing plasticity. The results on Atari show that plasticity injection attains stronger performance compared to alternative methods while being computationally efficient.

## C.2 Benchmark

**Meta-World**. Meta-World is an open-source benchmark for meta-reinforcement learning and multitask learning, comprising 50 distinct robotic manipulation tasks (Yu et al., 2020).

All tasks are executed by a simulated Sawyer robot, with the action space defined as a 2-tuple: the change in the 3D position of the end-effector, followed by a normalized torque applied to the gripper fingers.

The observation space has a consistent dimensionality of 39, although different dimensions correspond to various aspects of each task. Typically, the observation space is represented as a 6-tuple, including the 3D Cartesian position of the end-effector, a normalized measure of the gripper's openness, the 3D position and the quaternion of the first object, the 3D position and quaternion of the second object, all previous measurements within the environment, and the 3D position of the goal.

The reward function for all tasks is structured and multi-component, aiding in effective policy learning for each task component. With this design, the reward functions maintain a similar magnitudes across tasks, generally ranging between 0 and 10. The descriptions of the six tasks used in our experiments are listed below, and the appearance of these tasks is shown in Figure 5.

- **drawer-open**: Open a drawer, with randomized drawer positions.
- **drawer-close**: Push and close a drawer, with randomized drawer positions.
- **window-open**: Push and open a window, with randomized window positions.
- **window-close**: Push and close a window, with randomized window positions.
- **door-open**: Open a door with a revolving joint. Randomize door positions.
- **button-press-topdown**: Press a button from the top. Randomize button positions.

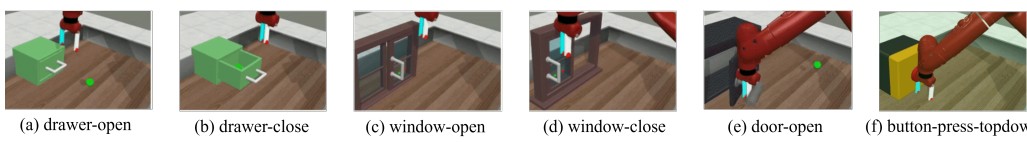

(a) drawer-open    (b) drawer-close    (c) window-open    (d) window-close    (e) door-open    (f) button-press-topdown

Figure 5: Tasks in the Meta-World benchmark used in our experiments.

**Atari**. Atari environments are simulated using the Arcade Learning Environment (ALE) (Bellemare et al., 2013) via the Stella emulator.

Each environment utilizes a subset of the full action space, which includes actions like NOOP, FIRE, UP, RIGHT, LEFT, DOWN, UPRIGHT, UPLEFT, DOWNRIGHT, DOWNLEFT, UPFIRE, RIGHTFIRE, LEFTFIRE, DOWNFIRE, UPRIGHTFIRE, UPLEFTFIRE, DOWNRIGHTFIRE, and DOWNLEFTFIRE. By default, most environments employ only a smaller subset of these actions, excluding those that have no effect on gameplay.

Observations in Atari environments are RGB images displayed to human players, with $obs\_type =$ "$rgb$", corresponding to an observation space defined as $Box(0, 255, (210, 160, 3), np.uint8)$.

The specific reward dynamics vary depending on the environment and are typically detailed in the game's manual.

The descriptions of the four games used in our experiments are listed below (Foundation, 2024), and the appearance of these games is shown in Figure 6.

- **Bowling**: The goal is to score as many points as possible in a 10-frame game. Each frame allows up to two tries. Knocking down all pins on the first try is called a "strike", while doing so on the second try is a "spare". Failing to knock down all pins in two attempts results in an "open" frame.
- **Pong**: You control the right paddle and compete against the computer-controlled left paddle. The objective is to deflect the ball away from your goal and into the opponent's goal.
- **BankHeist**: You play as a bank robber trying to rob as many banks as possible while avoiding the police in maze-like cities. You can destroy police cars using dynamite and refill your gas tank by entering new cities. Lives are lost if you run out of gas, are caught by the police, or run over your own dynamite.
- **Alien**: You are trapped in a maze-like spaceship with three aliens. Your goal is to destroy their eggs scattered throughout the ship while avoiding the aliens. You have a flamethrower to fend them off and can occasionally collect a power-up (pulsar) that temporarily enables you to kill aliens.

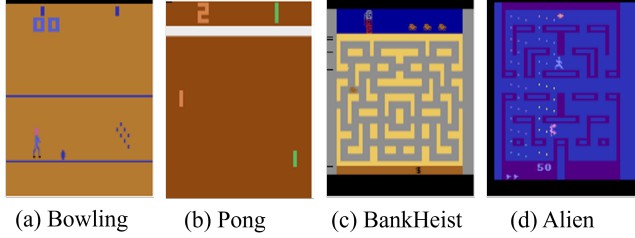

(a) Bowling    (b) Pong    (c) BankHeist    (d) Alien

Figure 6: Games in the Atari benchmark used in our experiments.

### C.3 Experiment setting

For all experiments, we utilize the open-source PyTorch implementation of Soft Actor-Critic (SAC) provided by CleanRL (Huang et al., 2022) on a single RTX2080Ti GPU. CleanRL is a Deep Reinforcement Learning library that offers high-quality, single-file implementations with research-friendly features. The code is both clean and straightforward, and we adhere to the configurations provided by CleanRL. During training, we employ an $\epsilon$-greedy exploration policy at the start, setting $\epsilon = 1$ for the first $10^4$ time steps to promote exploration. The environment is wrapped using Gym wrappers to facilitate experimentation. For the Meta-World benchmark, we utilize the RecordEpisodeStatistics wrapper to gather episode statistics. For the Atari benchmark, in addition to RecordEpisodeStatistics, we preprocess the $210 \times 160$ pixel images by downsampling them to $84 \times 84$ using bilinear interpolation, converting the RGB images to the YUV format, and using only the grayscale channel. Additionally, we set a maximum limit on the number of noop and skip steps to standardize the exploration.

Regarding network architecture, we use the same actor and critic networks for all tasks within the same benchmark to ensure consistency. For the Meta-World benchmark, we employ a neural network

comprising four fully connected layers, of which the hidden size is [768, 768, 768]. For the Atari benchmark, we use a convolutional neural network (CNN) with three convolutional layers featuring 32, 64, and 64 channels, respectively, followed by three fully connected layers, of which the hidden size is [768, 768].

To reduce randomness and enhance the reliability of our results, we train each agent using three random seeds. Additional hyper-parameters for the SAC algorithm applied in the Meta-World and Atari benchmarks are detailed in Table 6.

Table 6: Hyper-parameters of SAC in our experiments.

| Parameters | Values for Meta-World | Values for Atari |
|---|---|---|
| Initial collect steps | 10000 | 20000 |
| Discount factor | 0.99 | 0.99 |
| Training environment steps | $10^6$ | $1.5 \times 10^6, 3 \times 10^6$ |
| Testing environment steps | $10^5$ | $10^5$ |
| Replay buffer size | $10^6$ | $2 \times 10^5$ |
| Updates per environment step (Replay Ratio) | 2 | 4 |
| Target network update period | 1 | 8000 |
| Target smoothing coefficient | 0.005 | 1 |
| Optimizer | Adam | Adam |
| Policy learning rate | $3 \times 10^{-4}$ | $10^{-4}$ |
| Q-value learning rate | $10^{-3}$ | $10^{-4}$ |
| Minibatch size | 256 | 64 |
| Alpha | 0.2 | 0.2 |
| Autotune | True | True |
| Average environment steps of success rate | 10 | - |
| Stable threshold to finish training | 0.9 | - |
| Replay interval | 10 | 10 |
| No-op max | - | 30 |
| Target entropy scale | - | 0.89 |
| Storing experience size | $10^5$ | $10^5$ |

## C.4 Metrics

For the Meta-World benchmark, the average success rate is computed over 20 episodes. For the Atari benchmark, the success rate is replaced by the return of each episode. We normalize the return for each game to obtain summary statistics across games, as follows:

$$R = \frac{r_{agent} - r_{random}}{r_{human} - r_{random}}, \tag{12}$$

where $r_{agent}$ represents the average return evaluated over $10^5$ steps, the random score $r_{random}$ and human score $r_{human}$ are consistent with those used by Mnih et al. (2015), as detailed in Table 7.

Table 7: Normalization scores of Atari games.

| games | $r_{random}$ | $r_{human}$ |
|---|---|---|
| Bowling | 23.1 | 154.8 |
| Pong | -20.7 | 9.3 |
| BankHeist | 14.2 | 734.4 |
| Alien | 227.5 | 6875 |

For the Atari benchmark tasks, the overall performance is evaluated by Average Return (AR), which is analogous to ASR in the Meta-World benchmark. It is calculated as follows:

$$AR = \frac{1}{k} \sum_{i=1}^{k} \frac{1}{i} \sum_{i \geq j} R_{i,j}, \tag{13}$$

where $R_{i,j}$ represents the average return evaluated on the $j$-th task after completing the learning of the $i$-th task ($i \geq j$), and $k$ represents the number of tasks. A higher AR indicates better performance in balancing stability and plasticity.

## C.5    RL Skill Neurons

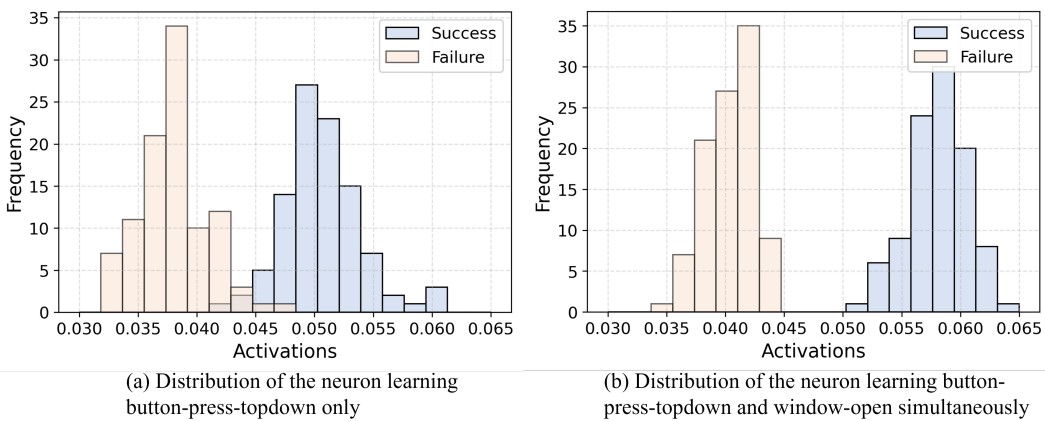

(a) Distribution of the neuron learning button-press-topdown only

(b) Distribution of the neuron learning button-press-topdown and window-open simultaneously

Figure 7: Distribution histogram of the activations of a neuron in two learning settings.

To validate the existence of RL skill neurons in sequential task learning instead of single task learning, we conduct an additional analysis comparing the activation distributions of neurons when learning button-press-topdown in isolation versus learning button-press-topdown and window-open simultaneously. As shown in Figure 7, the activation distribution of a representative neuron remains highly correlated with task success, regardless of whether it is learned in isolation or alongside another skill. This observation supports our hypothesis that skill-specific neurons retain their essential role even in a sequential task learning scenario.

Additionally, we dig deeper into the identified RL skill neurons and separate them into general and specific skills. How to deeply investigate general skills is key for our future research. To explore this, we design an experiment to verify the existence of general and specific skills. After sequentially training on the button-press-topdown and window-open tasks, we identify the RL skill neurons associated with each task. We hypothesize that the intersection set represents general skill neurons, while the difference set represents specific skill neurons. To validate this hypothesis, we zero out the outputs of these neurons separately. The results in Table 8 show that when the outputs of the general skill neurons are zeroed out, the agent fails to complete both tasks. In contrast, when the outputs of task-specific neurons are zeroed out, the agent can't complete the corresponding task but is still able to complete the other task. This confirms the existence of both general and specific skills.

Table 8: Results of zeroing out the output of general of specific skill neurons.

| tasks | zero out the intersection set | zero out the difference set of button-press-topdown relative to window-open | zero out the difference set of window-open relative to button-press-topdown |
|---|---|---|---|
| button-press-topdown | 0 | 0.33 | 1.00 |
| window-open | 0 | 1.0 | 0.42 |

## C.6    Results of Vanilla SAC

To validate the effectiveness of NBSP, it is essential to first confirm whether the vanilla SAC algorithm can successfully solve each task individually. So we conducted experiments by training a vanilla SAC agent on all tasks in our experiment. The results, presented in Figure 8, demonstrate that the vanilla SAC algorithm successfully learns all tasks in our experiment. This confirms that the balance between stability and plasticity is not an artifact of modifications to the SAC algorithm itself but

rather a result of NBSP. Furthermore, the failure of other methods is not due to limitations of the SAC algorithm.

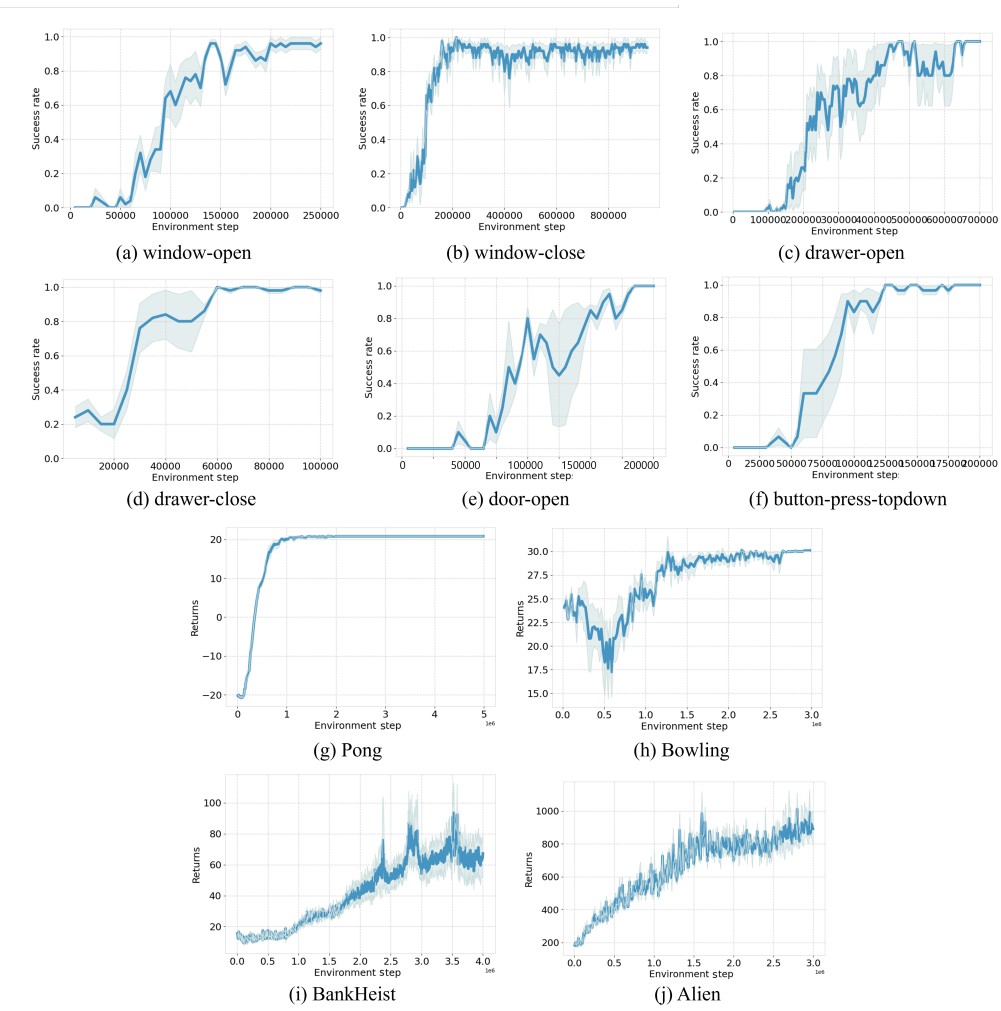

Figure 8: Training process of vanilla SAC on each individual task in our experiment.

## C.7   Results on the Meta-world Benchmark

The training process of the other four-tasks cycling task is shown in Figure 9, and those of the two-task cycling tasks are shown in Figure 10, Figure 11 and Figure 12 respectively. The same as found in Section 4.1, during the second cycle of learning the same task, the agent is able to master the task more rapidly.

## C.8   Ablation Study

The results of the ablation study on two critical components, gradient masking and experience replay techniques, are shown in Table 9 for the (window-open → window-close) cycling task and in Table 10 for the (drawer-open → drawer-close) cycling task. From these results, it is evident that both gradient masking and experience replay techniques independently contribute to improving the stability of the agent while maintain great plasticity. Furthermore, combining both techniques yields superior performance, demonstrating the enhanced effectiveness of their integration.

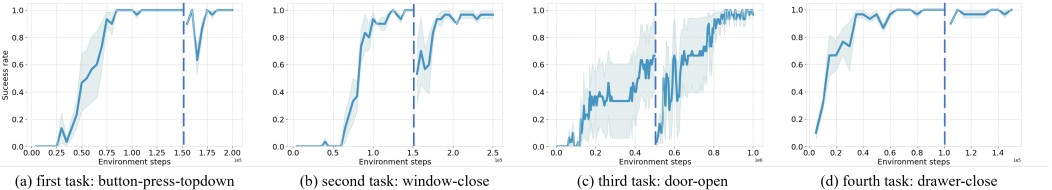

(a) first task: button-press-topdown   (b) second task: window-close   (c) third task: door-open   (d) fourth task: drawer-close

Figure 9: Training process of NBSP on (button-press-topdown → window-close → door-open → drawer-close) cycling task.

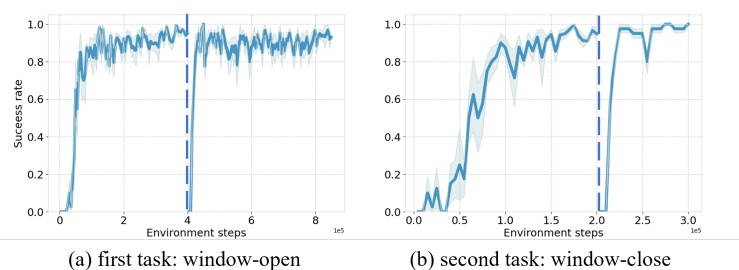

(a) first task: window-open   (b) second task: window-close

Figure 10: Training process of NBSP on (window-open → window-close) cycling task.

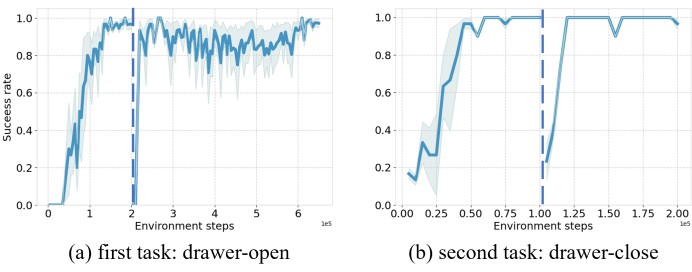

(a) first task: drawer-open   (b) second task: drawer-close

Figure 11: Training process of NBSP on (drawer-open → drawer-close) cycling task.

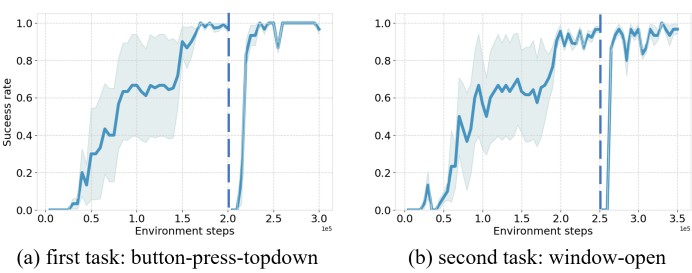

(a) first task: button-press-topdown   (b) second task: window-open

Figure 12: Training process of NBSP on (button-press-topdown → window-open) cycling task.

Table 9: Results of ablation study of gradient masking and experience replay techniques on (window-open → window-close) cycling task.

| Metrics | (button-press-topdown → window-open) | | | | |
| --- | --- | --- | --- | --- | --- |
| | vanilla SAC | only experience replay | only gradient masking | NBSP with hard gradient masking | NBSP |
| ASR ↑ | $0.63 \pm 0.02$ | $0.81 \pm 0.08$ | $0.78 \pm 0.11$ | $0.71 \pm 0.04$ | $\mathbf{0.90 \pm 0.04}$ |
| FM ↓ | $0.91 \pm 0.10$ | $0.41 \pm 0.13$ | $0.54 \pm 0.26$ | $0.54 \pm 0.13$ | $\mathbf{0.18 \pm 0.01}$ |
| FWT ↑ | $0.97 \pm 0.02$ | $0.96 \pm 0.01$ | $\mathbf{0.98 \pm 0.01}$ | $0.91 \pm 0.05$ | $\mathbf{0.96 \pm 0.02}$ |

Table 10: Results of ablation study of gradient masking and experience replay techniques on (drawer-open → drawer-close) cycling task.

| Metrics | (button-press-topdown → window-open) | | | | |
| --- | --- | --- | --- | --- | --- |
| | vanilla SAC | only experience replay | only gradient masking | NBSP with hard gradient masking | NBSP |
| ASR ↑ | $0.67 \pm 0.05$ | $0.78 \pm 0.04$ | $0.74 \pm 0.01$ | $0.59 \pm 0.16$ | $\mathbf{0.96 \pm 0.02}$ |
| FM ↓ | $0.78 \pm 0.10$ | $0.48 \pm 0.10$ | $0.64 \pm 0.01$ | $0.52 \pm 0.35$ | $\mathbf{0.07 \pm 0.06}$ |
| FWT ↑ | $0.94 \pm 0.04$ | $0.97 \pm 0.01$ | $\mathbf{0.98 \pm 0.02}$ | $0.82 \pm 0.21$ | $\mathbf{0.98 \pm 0.01}$ |

# D   Algorithm

The pseudo-code of the goal-oriented method to find RL skill neurons is presented in Algorithm 1. And the pseudo-code for SAC with NBSP is presented in Algorithm 2. Key differences from standard SAC are highlighted in blue. In addition to the extra input, two main modifications include the sampling process and the network update process.

# E   Limitation and Future Work

**Limitation**. While the proposed NBSP method effectively balances stability and plasticity in DRL, it does have a notable limitation. Specifically, the number of RL skill neurons must be manually determined and adjusted according to the complexity of the learning task, as there is no automatic mechanism for this selection. And our method currently faces challenges when applied to longer task sequences (e.g., 10+ tasks). One key limitation is the constraint imposed by the model scale, which inherently limits the number of skills it can learn. As the number of tasks increases, the overlap between skill neurons across different tasks may become significant. Consequently, applying a mask to protect RL skill neurons can restrict the learning of new tasks, making it difficult to scale without introducing interference with previously learned knowledge.

**Future work**. The neuron analysis introduced in this work offers a novel approach for identifying RL skill neurons, significantly enhancing the balance between stability and plasticity in DRL. The identification of RL skill neurons opens up several promising directions for future research and applications, such as: (1) Model Distillation: by focusing on RL skill neurons, it becomes possible to distill models by pruning less relevant neurons, leading to more efficient and compact models with minimal performance degradation. (2) Bias Control and Model Manipulation: RL skill neurons could be leveraged to control biases and modify model behaviors by selectively adjusting their activations. This approach could be particularly valuable in scenarios requiring specific outputs or behaviors.

While our current method may not yet fully address longer task sequences, it lays a strong foundation for future research. Moving forward, we aim to explore strategies to better leverage RL skill neurons for continual learning over an extended sequence of tasks. What's more, its applicable potential extends beyond DRL. It could also be adapted to other learning paradigms, such as supervised and unsupervised learning, to address similar stability-plasticity challenges. In future work, we plan to explore these extensions and verify their effectiveness across various domains.

**Algorithm 1** Procedure for Identifying RL Skill Neurons

**Input**: Initial average step $T_{avg}$, initial evaluation step $T$, initial proportion of RL skill neuron $m$, initial average activation $\overline{a}(\mathcal{N}) = 0$, initial average GM $\overline{q} = 0$, initial over-activation rate $R_{over} = 0$.

1: **for** each step $t$ **do**
2:     Compute activation $a(\mathcal{N}, t) \leftarrow \phi(\cdot)$
3:     Compute GM $q(t)$
4:     Compute average activation:

$$\overline{a}(\mathcal{N}) = \overline{a}(\mathcal{N}) + \frac{1}{T_{avg}} a(\mathcal{N}, t).$$

5:     Compute average GM:

$$\overline{q} = \overline{q} + \frac{1}{T_{avg}} q(t).$$

6: **end for**
7: **for** each step $t$ **do**
8:     Compute activation $a(\mathcal{N}, t) \leftarrow \phi(\cdot)$
9:     Compute GM $q(t)$
10:     Capture association:

$$R_{over} = R_{over} + \frac{1}{T} 1_{[1_{[a(\mathcal{N}, t) > \overline{a}(\mathcal{N})]} = 1_{[q(t) > \overline{q}]}]}$$

11: **end for**
12: Derive scores $Score$ for each neuron:

$$Score(\mathcal{N}) = max(R_{over}(\mathcal{N}), 1 - R_{over}(\mathcal{N}))$$

13: Identify the top-performing neurons as RL skill neurons:

$$\mathcal{N}_{RL\ skill} = \tau_m(Score(\mathcal{N}))$$

**Algorithm 2** Neuron-level Balance between Stability and Plasticity (NBSP) Applied in SAC

Initialize policy parameters $\theta$, Q-function parameters $\phi_1$, $\phi_2$, and target Q-function parameters $\phi_1'$, $\phi_2'$
Initialize empty replay buffer $\mathcal{D}$
Initialize replay interval $k$
**Input: Replay buffer $\mathcal{D}_{\mathbf{pre}}$, mask of the policy $\mathbf{mask}_\theta$ and mask of the Q-function parameters $\mathbf{mask}_{\phi_1}, \mathbf{mask}_{\phi_2}$**

1: **for** each task **do**
2:     **for** each iteration **do**
3:         **for** each environment step **do**
4:             Sample action $a_t \sim \pi_\theta(a_t|s_t)$
5:             Execute action $a_t$ and observe reward $r_t$ and next state $s_{t+1}$
6:             Store $(s_t, a_t, r_t, s_{t+1})$ in replay buffer $\mathcal{D}$
7:         **end for**
8:         **for** each gradient step **do**
9:             **if step $\equiv 0 \pmod{\mathbf{k}}$ then Sample batch of transitions $(\mathbf{s_i, a_i, r_i, s_{i+1}})$ from $\mathcal{D}_{\mathbf{pre}}$**
10:            **else** Sample batch of transitions $(s_i, a_i, r_i, s_{i+1})$ from $\mathcal{D}$
11:            **end if**
12:            Compute target value:

$$y_i = r_i + \gamma \left( \min_{j=1,2} Q_{\phi_j'}(s_{i+1}, \tilde{a}_{i+1}) - \alpha \log \pi_\theta(\tilde{a}_{i+1}|s_{i+1}) \right), where\ \tilde{a}_{i+1} \sim \pi_\theta(\cdot|s_{i+1})$$

13:            Update Q-functions by one step of gradient descent with mask:

$$\phi_j \leftarrow \phi_j - \lambda_Q \mathbf{mask}_{\phi_j} \nabla_{\phi_j} \frac{1}{N} \sum_i \left( Q_{\phi_j}(s_i, a_i) - y_i \right)^2 \quad \text{for } j = 1, 2$$

14:            Update policy by one step of gradient ascent with mask:

$$\theta \leftarrow \theta + \lambda_\pi \mathbf{mask}_\theta \nabla_\theta \frac{1}{N} \sum_i \left( \alpha \log \pi_\theta(a_i|s_i) - \min_{j=1,2} Q_{\phi_j}(s_i, a_i) \right)$$

15:            Update temperature $\alpha$ by one step of gradient descent:

$$\alpha \leftarrow \alpha - \lambda_\alpha \nabla_\alpha \frac{1}{N} \sum_i \left( -\alpha \log \pi_\theta(a_i|s_i) - \alpha \bar{\mathcal{H}} \right)$$

16:            Update target Q-function parameters:

$$\phi_j' \leftarrow \tau \phi_j + (1 - \tau)\phi_j' \quad \text{for } j = 1, 2$$

17:         **end for**
18:     **end for**
19:     **Select RL skill neurons $\{\mathcal{N}_{\mathbf{RL\,skill}}\}$ according to Algorithm 1**
20:     **Update $\mathbf{mask}_{\phi_1}, \mathbf{mask}_{\phi_2}$ and $\mathbf{mask}_\theta$:**

$$mask(\mathcal{N}) = \begin{cases} \alpha(1 - Score(\mathcal{N})) & \textbf{if } \mathcal{N} \in \mathcal{N}_{RL\,skill} \\ 1 & \textbf{if } \mathcal{N} \notin \mathcal{N}_{RL\,skill} \end{cases}$$

21:     **Store part of $\mathcal{D}$ into $\mathcal{D}_{\mathbf{pre}}$**
22: **end for**

