# OpenReview forum: "NBSP: A Neuron-Level Framework for Balancing Stability and Plasticity in Deep Reinforcement Learning"
_NeurIPS.cc/2025/Conference — Submitted to NeurIPS 2025_

### Official Review · Reviewer_AmTw · 2025-06-21

**Clarity:** 4
**Significance:** 4
**Originality:** 4
**Rating:** 5
**Confidence:** 5

**Summary:**

This paper addresses a fundamental problem in deep reinforcement learning, namely the balance between stability and plasticity. The authors propose a methodology to identify critical neurons, whose updates (at least partially) are controlled through a masking mechanism.  Once these RL skills neurons are identified, a replay buffer further enhances the stability of the overall method. The method is extensively evaluated and benchmarked against several recent baselines. The results are very competitive, convincingly showcasing the effectiveness of the proposed methodology.

**Questions:**

- In Sec. 3.3, the fact that $\alpha$ is set to 0.2 appears to be a rather empirical choice. How was this value selected, and what's the impact on the performance? I would have expected to see a sensitivity study on this parameter (and I would appreciate it if the authors could show this with some additional experiments). Also, to avoid any confusion, the ablation on the proportion of RL skill neurons (where I guess the sweet spot value 0.2 refers to the top m% indicated at p. 5, l. 187) should make it clear that it is *not* about the $\alpha$ parameter used in Eq. 7, but rather on the top-m value. The fact that the same value of 0.2 is used for both parameters may be confusing.
- From the description of the method, it's not clear what happens if the same neuron is identified as an RL skill neuron for multiple tasks. Is this a possibility? If so, how is this handled in the method (particularly wrt Eq.s 4, 6, and 7)?
- When describing the experimental setup in Sec. 4, the authors explain that repeating each task twice within a cycling sequence avoids the influence of task order. As far as I can see, the task order is, however, fixed in the experiment. Could the authors elaborate further as to why repeating tasks twice avoids the influence of task order?
- Connected to the previous question, it would be interesting to see a set of experiments where the task order is indeed *randomly shuffled*, to see if there is indeed any task order effect. The fixed sequence detailed in App. C.2 suggests that the order may play a role (e.g., drawer-open and drawer-close, which essentially require the same actions but in reverse order, come one after the other; same for window-open and window-close). Would it be possible to see such experiments to rule out any task order effect?
- While results clearly show that the proposed method outperforms the baselines, the fact that only 3 seeds are used (which I understand may be due to the high computational costs of DRL experiments) may call for more cautious statements (e.g., in Sec. 4.1 "significantly outperforms ", "substantially higher than other baselines", etc.). In the absence of statistical evidence, these statements may be read as overly bold and perhaps should be rephrased.
- Several important aspects that normally appear in the main text (incl. the related work) are moved to the Appendix. While I understand the rationale behind that, perhaps a better balance between the different sections (e.g., considering the ablations) should be sought to have a self-contained main text.
- Other works have proposed neuron-level plastic NNs models, see e.g. the node encoding in https://doi.org/10.1016/S0893-6080(00)00032-0 and the more recent NcHL approach from https://arxiv.org/abs/2403.12076. Could the authors elaborate in the related work section how does the present submission relate to these works?
- To allow reproducibilty, the codebase of this work should be made publicly available.

Minor
- L. 292, missing space between "Table 2." and "From"

**Ethical Concerns:**

["NO or VERY MINOR ethics concerns only"]

**Final Justification:**

I confirm my score.

**Limitations:**

The authors discuss the limitations in Appendix E. I would suggest moving this part to the main paper.

**Paper Formatting Concerns:**

The paper appears well-formatted.

**Quality:**

4

**Strengths And Weaknesses:**

Strenghts
- Excellent presentation and quality of the overall paper
- Well-thought-out methodology
- Strong evaluation, including multiple baselines and complete ablations
- Strong numerical results

Weaknesses
- Some hyperparameters are chosen empirically
- A study on task order is missing (see Questions)
- Some parts of the Appendix (related work, limitations) should be moved to the main text

---

> ### Author Rebuttal · Authors · 2025-07-30
>
> We appreciate your valuable feedback on the hyper-parameter, task order, writing, and your acknowledgement of presentation, novelty, and experimental validation of our paper. We will address each of your comments and concerns in the following responses.
>
>   Q1：the chosen of hyperparameters
>
>   A1: Thanks for the question. We present sensitivity analysis of the hyperparameters as following and provide a practical guideline for each one:
>
>   - $m$: As discussed in lines 331-345 in the paper, the performance improves as $m$ increases, but it begins to decline after a certain threshold. The results in Figure 4 demonstrate that **NBSP consistently delivers great ASR improvements across a broad range of $m$ values (0.15 - 0.3)**, indicating that **its selection is not overly sensitive within a practical operational range**. Setting $m$ within this range yields strong performance gains across different tasks and benchmarks in our experiments.
>   - $\alpha$:
>     We vary $\alpha$ from 0.1 to 1.0. As $\alpha$ decreases, FM consistently improves, indicating better stability, while FWT remains relatively stable, suggesting that **plasticity is not highly sensitive to $\alpha$. Notably, ASR performance is strong as long as $\alpha < 0.3$, Overall, NBSP is robust to the choice of $\alpha$ within this range**.
> |$\alpha$|ASR $\uparrow$|FM $\downarrow$|FWT $\uparrow$|
> |:-|:-|:-|:-|
> |0.1|0.93±0.04|0.07±0.10|0.96±0.01|
> |0.2|0.95±0.05|0.08±0.12|0.98±0.01|
> |0.3|0.91±0.07|0.13±0.16|0.98±0.01|
> |0.5|0.85±0.01|0.33±0.00|0.98±0.01|
> |1.0|0.81±0.06|0.48±0.15|0.98±0.01|
>   - $|D{pre}|$:
>     We vary buffer size ranging from 1e2 to 1e6. When the buffer size is too small, previous task information cannot be fully stored, leading to stability loss (high FM). However, **when $|D{pre}|$ reaches around 1e5, NBSP performs well and remains insensitive to buffer size beyond this point**.
> |$D_{pre}$|ASR $\uparrow$|FM $\downarrow$|FWT $\uparrow$|
> |:-|:-|:-|:-|
> |1e2|0.62±0.01|0.99±0.01|0.99±0.01|
> |1e3|0.62±0.01|0.99±0.01|0.98±0.01|
> |1e4|0.74±0.09|0.67±0.21|0.98±0.01|
> |1e5|0.95±0.05|0.08±0.12|0.98±0.01|
> |1e6|0.93±0.04|0.13±0.13|0.99±0.01|
>   - $k$:
>     We test values of $k$ ranging from 2 to 100. When $k$ is small, frequent replay of previous experiences enhances stability but reduces plasticity (low FWT). In contrast, past experiences are underutilized, weakening stability. **When $k$ is within the range of 5-13, NBSP performs well, demonstrating insensitivity to variations in $k$ in this range**.
> |$k$|ASR $\uparrow$|FM $\downarrow$|FWT $\uparrow$|
> |:-|:-|:-|:-|
> |2|0.62±0.01|0.02±0.02|0.50±0.00|
> |5|0.95±0.04|0.07±0.09|0.97±0.02|
> |10|0.95±0.05|0.08±0.12|0.98±0.01|
> |13|0.94±0.04|0.11±0.09|0.98±0.01|
> |20|0.89±0.06|0.21±0.13|0.98±0.01|
> |100|0.66±0.01|0.90±0.05|0.99±0.01|
>
>   Q2: $\alpha$ and $m$
>
>   A2: Thank you for the insightful comment. $\alpha$ controls the strength of the adaptive gradient masking by scaling the mask value applied to RL skill neurons. A smaller $\alpha$ imposes stronger restrictions on these neurons, enhancing stability but potentially impairing plasticity. When $\alpha = 0$, the method degenerates into hard gradient masking and loses its adaptive property. We set $\alpha = 0.2$ based on its empirical performance, as it yields the best results in our experiments. To evaluate its impact, we have conducted an sensitive study, which is shown in the above A1.
>
>   We also appreciate your feedback regarding the confusion between the values of $\alpha$ and $m$. To clarify, t**he value of 0.2 used for $m$ refers to the top-m% of RL skill neurons, and is based on the highest ASR value observed**, as shown in Figure 4. We will make this distinction clearer in the revised manuscript.
>
>   Q3: same neurons is identified as an RL skill neuron for multiple tasks
>
>   A3: Thank you for the insightful question. We investigate the proportion of neurons identified as RL skill neurons by two tasks to the total number of RL skill neurons, as shown in the table below. The results confirm that some neurons are indeed identified as RL skill neurons for multiple tasks. When the same neuron is identified as an RL skill neuron for multiple tasks, we handle this by **averaging its mask values**. The mask value is adaptively computed based on the score of neuron for each task, ensuring that the influence of the neuron is appropriately adjusted across tasks.
>
>   |   tasks         | window-open & window-close | window-open & button-press-topdown | button-press-topdown & window-close |
>   | :--------: | :------------------------: | :--------------------------------: | :---------------------------------: |
>   | proportion |            26%             |                25%                 |                 29%                 |
>
>   Q4: task order
>
>   A4: Thanks for the insightful question. In our experimental setup, **we repeat each task twice within a cycling sequence to mitigate the potential influence of task order**. For example, in a two-task cycling sequence (button-press-topdown → window-open), repeating these tasks results in the sequence (button-press-topdown → window-open → button-press-topdown → window-open). This sequence contains two occurrences of the subsequence (button-press-topdown → window-open) and one occurrence of (window-open → button-press-topdown), which helps mitigate the impact of task order. However, we acknowledge that this approach does not fully eliminate the influence of task order, and we will revise the description to clarify this point.
>
>   To further investigate the effect of task order, we conduct experiments with randomized task order. The results, shown in the table below, indicate that **task order does affect performance, particularly in terms of stability. However, the impact is modest, and NBSP still performs well in balancing stability and plasticity regardless of the task order. Determining task order presents a promising future direction, where task difficulty, diversity, and coherency might be taken into account.**
>
>   |       cycling sequential tasks       |     ASR$\uparrow$     |     FM$\downarrow$     |     FWT$\uparrow$     |
>   | :---------------------------------- | :--------- | :---------| :---------|
>   | (window-open → button-press-topdown) | 0.9 ± 0.08  | 0.17 ± 0.13 | 0.95 ± 0.02 |
>   | (button-press-topdown → window-open) | 0.95  0.05 | 0.08 ± 0.12 | 0.98 ± 0.01 |
>   | (drawer-open → drawer-close)   | 0.96 ± 0.02   |    0.07 ± 0.06         |   0.98 ± 0.01   |
>   | (drawer-close → drawer-open)  | 0.92 ± 0.05   |    0.12 ± 0.12         |   0.97 ± 0.01   |
>
>   Q5: statements may be read as as overly bold
>
>   A5: Thank you for pointing out the issue with the statements regarding experimental results, particularly given the limited number of seeds. We agree that terms such as "significantly" and "substantially" could be misleading in the absence of statistical validation. We will remove these terms and rephrase the statements to reflect a more cautious interpretation of the results in the revised manuscript.
>
>   Q6: important aspects in the appendix
>
>   A6: Thank you for your suggestion. We will aim for a better balance between the sections by including a more refined analysis of the ablation experiments and moving important aspects, such as related work and limitation, into the main text. Additionally, we will consider adjusting the structure to ensure the paper is self-contained. Should the paper be accepted, we'll leverage the additional page to accommodate these changes seamlessly.
>
>   Q7: other neuron-level plastic NNs models
>
>   A7: Thanks for the suggestion. We will include the discussion and comparison in the revised manuscript by:
>
>   "Neural network plasticity is crucial for intelligent systems to adapt and learn complex behaviors. Foundational research has extensively explored integrating and managing this plasticity at the neuronal or synaptic level. Floreano et al.[1] notably demonstrated the significance of online self-organizing synapses in evolutionary robotics, providing robust evidence for dynamic neuronal adaptability. More recently, Ferigo et al. [2] enhanced neuron-level plasticity's efficiency with their Neuron-centric Hebbian Learning (NcHL) model. Their innovation of elevating Hebbian rule parameters specific from neurons rather than synapses highlights the potential for streamlined plasticity mechanisms.
>
>   Our work aligns with Floreano et al.'s pursuit of neuron-level adaptability for robust behavior, and shares conceptual commonalities with Ferigo et al.'s neuron-centric approach to plasticity control. Our distinct contributions include: (1) applying this neuron-level perspective to address the stability-plasticity dilemma in RL; and (2) introducing the novel concept of RL skill neurons to precisely identify neurons critical for mastering task-specific skills."
>
>   Q8: the codebase of this work
>
>   A8: Thank you for the suggestion. We have provided our code in the supplementary material and will make it publicly available in the near future to ensure reproducibility.
>
>   Q9: missing space between "Table 2." and "From"
>
>   A9: Thank you for pointing it out. We will fix it in the revised manuscript.
>
>
>
>   [1] Floreano D, et al. "Evolutionary robots with on-line self-organization and behavioral fitness." (2000).
>
>   [2] Ferigo A, et al. "Neuron-centric hebbian learning"(2024).

---

> > ### Comment · Reviewer_AmTw · 2025-08-01
> >
> > I thank the authors for addressing all my concerns. I am satisfied with the provided replies, and I confirm my rating (5).
> >
> > All the additional ablations, results, and discussions indicated in this rebuttal should be included in the final version of the manuscript.

---

> > > ### Author Response · Authors · 2025-08-05
> > >
> > > Thanks a lot for your efforts in reviewing this paper and proposing these valuable insights and suggestions. We will include all the additional ablation results and discussions in the revised manuscript as well.

---

### Official Review · Reviewer_4Ero · 2025-06-27

**Clarity:** 2
**Significance:** 2
**Originality:** 2
**Rating:** 4
**Confidence:** 5

**Summary:**

The paper introduces NBSP, a neuron-level continual deep-RL framework tackling both stability and plasticity. It consists of three components: 1) activation-based selection of "skill neurons", 2) gradient masking based on neuron scores, and 3) a memory replay mechanism. NBSP reduces forgetting by adaptively masking the gradients of neurons that contribute to high performance, combined with periodically replaying experiences from previous tasks. By performing regular updates on neurons that do not contribute to task performance, NBSP maintains plasticity on new tasks. The authors combine their method with SAC and evaluate on cyclical versions of MetaWorld and Atari. They provide ablation studies examining the importance of different components in NBSP.

**Questions:**

- How are the samples for experience replay selected? I couldn't find this specified in the paper.
- Are skill neurons concentrated more in the earlier or in the later layers of the network?
- Would the algorithm also work for on-policy algorithms such as PPO?
- How much slower compared to regular SAC is NBSP, given that it has to compute and maintain scores with each forward pass?
- How would the author set the hyperparameter $m$ related to network size? E.g., select 20% of parameters as $m$?
- How sensitive is NBSP for its main hyperparameters $m$, $\alpha$, and $k$?

**Ethical Concerns:**

["NO or VERY MINOR ethics concerns only"]

**Final Justification:**

The authors provided extensive experiments and clarifications in response to my review, addressing all of my concerns. Therefore, I have raised my score to 4.

**Limitations:**

Limitations are included but only in the Appendix.

**Quality:**

2

**Strengths And Weaknesses:**

Overall, I find the paper to be well-motivated and the proposed method to be sensible. However, I do have severe questions about evaluation, particularly the selected benchmark methods. Without including more methods that are a) specifically targeting the stability-plasticity trade-off and b) shown to work well in continual RL, I cannot recommend acceptance. If the authors were to include these baselines and show favorable performance of NBSP, I would raise my score.

## Strengths

- The work is well-motivated. It makes sense to look at neuron-level activations in the context of continual RL.
- The ablation studies are well-designed and provide meaningful insights.
- The authors evaluate on continuous as well as discrete control tasks.
- Code is provided, ensuring reproducibility of results.

## Weaknesses

MAJOR:

- The baselines are weak, making the evaluation misleading. Except for ANCL and CoTASP, the authors only benchmark against methods that target either plasticity or stability, but not both. ANCL and CoTASP do not use experience replay, which gives NBSP a big advantage in the off-policy setting. Several baselines that address stability and plasticity are missing: Shrink & Perturb [1], UPGD [2], and NE [3].
- All three of the methods that I mentioned are not even mentioned in the related work, despite being either very well established (Shrink & Perturb), clearly solving the same stability-plasticity trade-off (UPGD), or being methodologically very similar (NE).
- The algorithm introduces a lot of hyperparamters which seem to be hard to tune, particularly for larger agents: $m$ for the top-m activation selection, $\alpha$ for mask generation, the size of the experience buffer $|D_{pre}|$, how often to replay experience from previous tasks $k$.
- The proposed framework is applied only to a single RL algorithm (SAC).

MINOR:

- I find the writing in the paper to be not very clear, and some details are omitted. For example, how transitions for $D_{pre}$ are selected.
- Some of the claims seem a bit exaggerated given the limited evaluation, e.g., "NBSP exhibits excellent generalization in balance stability and plasticity across different benchmarks".
- Limitations are hidden in the Appendix.
- Not all of the NBSP hyperparameters seem to be specified in Table 6 of the appendix.
- No hyperparameter sensitivity analysis.

[1] Ash, Jordan, and Ryan P. Adams. "On warm-starting neural network training." (2020).

[2] Elsayed, Mohamed, and A. Rupam Mahmood. "Addressing loss of plasticity and catastrophic forgetting in continual learning." (2024).

[3] Liu, Jiashun, et al. "Neuroplastic Expansion in Deep Reinforcement Learning." (2024).

---

> ### Author Rebuttal · Authors · 2025-07-30
>
> We appreciate your valuable feedback on the baseline, related work, hyper-parameter, algorithm, writing, and training time of our paper. We will carefully address each of your comments below and also in our revised manuscript.
>
>   Q1: baselines
>
>   A1: We acknowledge that some of the baselines primarily focus on either stability or plasticity. We would like to highlight that **this presents one of the advantages of our approach, which explicitly targets a balance of both**.
>
>   To ensure fair comparison, we add experience replay to ANCL and CoTASP on the cycling task sequence (button-press-topdown → window-open). As shown in the following table, **ANCL with experience replay heavily improves stability (FM from 0.95 to 0.42), while CoTASP with experience replay gets no substantial performance gain**, for the characteristics of tasks differ from the knowledge distribution captured by the existing dictionary, preventing the generation of effective sparse prompts. **And NBSP achieves better performance on all metrics**.
> |method|ASR $\uparrow$|FM $\downarrow$|FWT $\uparrow$|
> |:-|:-|:-|:-|
> |ANCL|0.61±0.01|0.95±0.05|0.95±0.03|
> |ANCL+experience replay|0.83±0.07|0.42±0.18|0.97±0.02|
> |CoTASP|0.03±0.00|0.01±0.00|0.04±0.01|
> |CoTASP+experience replay|0.03±0.01|0.01±0.01|0.05±0.01|
> |NBSP|0.95±0.05|0.08±0.12|0.98±0.01|
>
>   Additionally, we further compare NBSP with **Shrink & Perturb, NE, and UPGD**, which all aim to balance stability and plasticity. Experience replay is added for fair comparison as well. As shown below, Shrink & Perturb maintains high plasticity (high FWT) but suffers from significant stability loss (high FM), partially mitigated by experience replay. NE behaves similarly, failing to maintain stability. UPGD performs worst, struggling with both stability and plasticity. In comparison, **NBSP consistently outperforms these baselines in balancing both stability and plasticity, highlighting its potential**.
> |method|ASR $\uparrow$|FM $\downarrow$|FWT $\uparrow$|
> |:-|:-|:-|:-|
> |Shrink&Perturb|0.64±0.03|0.93±0.07|0.98±0.02|
> |Shrink&Perturb+experience replay|0.72±0.04|0.70±0.13|0.97±0.02|
> |NE|0.60±0.03|1.00±0.00|0.94±0.05|
> |NE+experience replay|0.71±0.03|0.73±0.10|0.96±0.01|
> |UPGD|0.39±0.03|0.82±0.05|0.54±0.07|
> |UPGD+experience replay|0.51±0.06|0.68±0.14|0.71±0.13|
> |NBSP|0.95±0.05|0.08±0.12|0.98±0.01|
>
>   Q2: related work
>
>   A2: We enrich the related work section by:
>
>   "Shrink & Perturb efficiently updates model without sacrificing generalization by shrinking the weights and adding small parameter noise. It risks stability loss and uniform parameter treatment also lacks fine-grained control over critical parameters. UPGD balances stability and plasticity by applying varied update sizes based on unit utility, yet its step-wise variable updates could slow down the training process. NE dynamically adapts network topology via gradient-based neuron growth and pruning. However, it overlooks the link between a neuron and the task's goal in RL and its topology adjustments are time-consuming. In contrast, NBSP moves beyond static methods by identifying and leveraging the correlation between a neuron's activation and the specific objectives of the RL task. Furthermore, our approach identify neurons after the training process, without impacting the training process itself."
>
>   Q3: hyper-parameters
>
>   A3: Thanks for the question. **we investigate the sensitivity of these parameters on a cycling task sequence (button-press-topdown → window-open)**. The results demonstrate that NBSP performs well within certain robust ranges of hyper-parameters, making it easier to tune.
>   - $m$: As discussed in lines 331-345 in the paper, the performance improves as $m$ increases, but it begins to decline after a certain threshold. The results in Figure 4 demonstrate that **NBSP consistently delivers great ASR improvements across a broad range of $m$ values (0.15 - 0.3)**, indicating that **its selection is not overly sensitive within a practical operational range**. Setting $m$ within this range yields strong performance gains across different tasks and benchmarks in our experiments.
>   - $\alpha$:
>     We vary $\alpha$ from 0.1 to 1.0. As $\alpha$ decreases, FM consistently improves, indicating better stability, while FWT remains relatively stable, suggesting that **plasticity is not highly sensitive to $\alpha$. Notably, ASR performance is strong as long as $\alpha < 0.3$, Overall, NBSP is robust to the choice of $\alpha$ within this range**.
> |$\alpha$|ASR $\uparrow$|FM $\downarrow$|FWT $\uparrow$|
> |:-|:-|:-|:-|
> |0.1|0.93±0.04|0.07±0.10|0.96±0.01|
> |0.2|0.95±0.05|0.08±0.12|0.98±0.01|
> |0.3|0.91±0.07|0.13±0.16|0.98±0.01|
> |0.5|0.85±0.01|0.33±0.00|0.98±0.01|
> |1.0|0.81±0.06|0.48±0.15|0.98±0.01|
>   - $|D{pre}|$:
>     We vary buffer size ranging from 1e2 to 1e6. When the buffer size is too small, previous task information cannot be fully stored, leading to stability loss (high FM). However, **when $|D{pre}|$ reaches around 1e5, NBSP performs well and remains insensitive to buffer size beyond this point**.
> |$D_{pre}$|ASR $\uparrow$|FM $\downarrow$|FWT $\uparrow$|
> |:-|:-|:-|:-|
> |1e2|0.62±0.01|0.99±0.01|0.99±0.01|
> |1e3|0.62±0.01|0.99±0.01|0.98±0.01|
> |1e4|0.74±0.09|0.67±0.21|0.98±0.01|
> |1e5|0.95±0.05|0.08±0.12|0.98±0.01|
> |1e6|0.93±0.04|0.13±0.13|0.99±0.01|
>   - $k$:
>     We test values of $k$ ranging from 2 to 100. When $k$ is small, frequent replay of previous experiences enhances stability but reduces plasticity (low FWT). In contrast, past experiences are underutilized, weakening stability. **When $k$ is within the range of 5-13, NBSP performs well, demonstrating insensitivity to variations in $k$ in this range**.
> |$k$|ASR $\uparrow$|FM $\downarrow$|FWT $\uparrow$|
> |:-|:-|:-|:-|
> |2|0.62±0.01|0.02±0.02|0.50±0.00|
> |5|0.95±0.04|0.07±0.09|0.97±0.02|
> |10|0.95±0.05|0.08±0.12|0.98±0.01|
> |13|0.94±0.04|0.11±0.09|0.98±0.01|
> |20|0.89±0.06|0.21±0.13|0.98±0.01|
> |100|0.66±0.01|0.90±0.05|0.99±0.01|
>
> Q4: single RL algorithm:
>
>   A4: Thanks for the insightful question. We further apply NBSP to PPO on a cycling task sequence (button-press-topdown→window-open). The results show that vanilla PPO performs worse than vanilla SAC in our setting, suffering from both stability and plasticity loss. **NBSP helps reduce FM, improving stability and achieving a better balance, as reflected by a higher ASR. However, the effect is less pronounced than that of SAC**. Potential reasons include:
>   1. **On-Policy Nature of PPO**: PPO is an on-policy algorithm, and cannot fully leverage the experience replay mechanism. While old experiences can still be sampled, they are more likely located outside the "trust region", leading to suboptimal updates.
>   2. **Differences in Exploration Mechanisms**: SAC incorporates an entropy regularization term in its objective function. When NBSP masks RL skill neurons, the entropy term of SAC helps maintain exploration in other neurons, without sacrificing too much plasticity. In contrast, PPO’s exploration is driven primarily by its stochastic policy and lacks explicit entropy constraint, making it more prone to instability if RL skill neurons are masked.
>
> |method|ASR $\uparrow$|FM $\downarrow$|FWT $\uparrow$|
> |:-|-|-|-|
> |vanilla PPO|0.40±0.04|0.82±0.18|0.66±0.12|
> |PPO with NBSP|0.49±0.06|0.58±0.09|0.67±0.11|
>
>   Q5: writing in the paper
>
>   A5: Thanks for pointing it out. $D{pre}$ stores the experience from previously encountered tasks. **After completing the learning of each task, we randomly sample experiences from the replay buffer for the current task and add them to $D{pre}$**. The sample size is determined as $|D_{pre}|/H$, where $|D_{pre}|$ denotes the buffer size, $H$ is the number of tasks.
>
>   Q6: some claims seem a bit exaggerated, limitations
>
>   A6: Thanks for pointing it out. We **will revise the writing in the next version to avoid exaggerated claims and include the limitation in the main text**. For example, we will modify the mentioned statement to "NBSP demonstrates generalization ability in balancing stability and plasticity across the Meta-World and Atari benchmarks."
>
>   Q7: lack of some hyper-parameters in the appendix
>
>   A7: Thanks for pointing it out. We will include Average steps $T_{avg}$ and Proportion of RL skill neurons $m$ in Table 6 in the revised manuscript.
>
>   Q8: samples of experience replay
>
>   A8: The samples for experience replay are randomly selected from the replay buffer.
>
>   Q9: distribution of the RL skill neurons
>
>   A9: We averaged the proportion of RL skill neurons in each layer across three tasks (button-press-topdown, window-open, window-close). The results in each layer, excluding the last one, are: 0.2%, 77.6%, and 22.2% in the first, second, and third layers, which indicates that **RL skill neurons are more concentrated in the middle layers of the network**.
>
>   Q10: training time
>
>   A10:  NBSP does not compute and maintain scores during training. **The neuron identification process is performed after completing each task, which takes approximately 10 minutes** on average across three tasks (button-press-topdown, window-open, window-close). After that, the new masks are set. **During training, the masks are only multiplied when computing weight updates, which takes 0.0017s for each update, and the additional computational overhead is negligible**.
>
>   Q11: the hyperparameter $m$ related to network size
>
>   A11: Thanks for the insightful question. We kindly refer to A3 for the impact of $m$. To ensure a fair comparison, the network size in our experiments is fixed, and we have not yet explored the effect of network size on the proportion of RL skill neurons. This presents a promising future direction, and we hypothesize that as the network size increases, the proportion per task may decrease. Regardless of model size, the skills required to master a task are finite. While encoding these necessary skills may involve a certain number of neurons, this proportion won't increase indefinitely as the model scales up.

---

> ### Author Response · Authors · 2025-08-05
>
> Thank you for your careful review of our rebuttal and for the insightful comments. And we would like to provide further details regarding the PPO implementation and the baselines. We will also include the discussions of the entropy bonus in the revised manuscript for better clarification.
>
> (1) PPO and Entropy Bonus: Thanks for raising the discussion on entropy-based exploration in PPO. We agree that this is an important technique in online RL and many standard PPO implementations, such as CleanRL, do include an entropy bonus, especially for discrete actions. However, we conducted our experiments in continuous-action environments, and followed the default setting of the PPO implementation in CleanRL for continuous action, which sets the entropy bonus coefficient as 0 (cleanrl/ppo_continuous_action.py).
>
> We suspect that there are two potential reasons for using entropy coefficient=0 for PPO with continuous actions:
> - Exploration Mechanism: In continuous action spaces, PPO typically outputs the mean and standard deviation of a Gaussian distribution. The standard deviation itself serves as an exploration mechanism, and its magnitude is learned and adjusted throughout training.
>
> - Entropy Calculation: For continuous actions, the entropy is usually derived from the Gaussian distribution, as implemented in CleanRL. As a result, the estimation of the entropy value might be less accurate, which depends on the quality of the Gaussian approximation of the action. In contrast, for discrete actions, we can directly estimate the entropy using the probabilities predicted for each action.
>
> To investigate the effect of the entropy bonus in our continuous-action setting, we further conduct an experiment on the cycling task sequence (button-press-topdown → window-open), using PPO with and without the entropy bonus. As shown in the following table, PPO without the entropy bonus performed slightly better. (Remark: we only experiment with entropy coeffient=0.01, and further tuning this hyperparameter may improve the performance of PPO with entropy)
>
>   | Method        | ASR $\uparrow$      | FM $\downarrow$       | FWT $\uparrow$      |
>   | :------------ | --------- | --------- | --------- |
>   | PPO + NBSP | 0.49±0.06 | 0.58±0.09 | 0.67±0.11 |
>   | PPO with entropy + NBSP | 0.44±0.04 | 0.69±0.07 | 0.60±0.16 |
>
> (2) Implementation Details of Additional Baselines (UPGD, Shrink & Perturb, NE)
>
> Thanks for the clarification question on the baselines.
>
> - UPGD: We use the official implementation provided by the authors of the UPGD paper: repo:mohmdelsayed/upgd.
>
> - Shrink & Perturb and NE: For these baselines, as no official code is provided in their original papers, we use the open-sourced implementations available on GitHub：
>   - Shrink & Perturb: repo:JordanAsh/warm_start
>   - NE: repo:torressliu/neuroplastic-expansion
>
> We will also include the references for the code repositories in the final version of our manuscript.
> Regarding the choice of hyperparameters, we used the default settings specified in the repositories. The key hyperparameters are listed in the following table for your kind reference.
>
> - Shrink & Perturb
>
>     | hyper-parameter | descripion          | value |
>     | --------------- | ------------------- | ----- |
>     | $\alpha$        | shrink coefficient  | 0.4   |
>     | $\sigma$        | perturb coefficient | 0.1   |
>
> - UPGD
>
>     | hyper-parameter | descripion                 | value |
>     | --------------- | -------------------------- | ----- |
>     | $\beta_u$       | utility trace decay factor | 0.999 |
>     | $\sigma$        | noise standard deviation   | 0.001 |
>     | $\lambda$       | weight decay               | 0.001 |
>
> - NE
>
>     | hyper-parameter | descripion                           | value |
>     | --------------- | ------------------------------------ | ----- |
>     | sparsity_stl    | initial sparsity for STL mode        | 0.8   |
>     | actor_sparsity  | target sparsity ratio for the actor  | 0.25  |
>     | critic_sparsity | target sparsity ratio for the critic | 0.25  |
>     | $\Delta T$      | grow interval                        | 10000 |e
>
> Thank you again for your constructive feedback. And we would like to provide more details if there is anything you would like to further discuss.

---

> > ### Comment · Reviewer_4Ero · 2025-08-05
> >
> > Thank you for the additional experiments and details. You have addressed my concerns satisfactorily. Therefore, I will raise my score to 4.

---

> > > ### Author Response · Authors · 2025-08-06
> > >
> > > Thanks a lot for your efforts in reviewing this paper and proposing these valuable insights and suggestions. We will include all the additional experiments and details in the revised manuscript as well.

---

### Official Review · Reviewer_yS8E · 2025-06-30

**Clarity:** 3
**Significance:** 2
**Originality:** 2
**Rating:** 5
**Confidence:** 4

**Summary:**

The paper proposes a strategy for balancing stability and plasticity in reinforcement learning. To support continous adaptation to new tasks, the authors propose to identify neurons that have an activation pattern which is correlated to task success. Applying gradient masking to these neurons supports preserving already learned skills. Meanwhile, the other neurons can freely adapt to new tasks. To further reinforce stability, experience replay, i.e. periodically revisiting past experience prevents excessive drift from previous knowledge.

**Questions:**

How does your method perform on more complex benchmarks?
Are there any theoretical explainations why it works?
Could you explain why you exactly combined those building blocks that you used?

**Ethical Concerns:**

["NO or VERY MINOR ethics concerns only"]

**Final Justification:**

In the rebuttal, the authors addressed all my questions convincingly. Therefore, I raised my score.

**Limitations:**

A short discussion of limitations in the main paper would be good. Currently only in the appendix.

**Paper Formatting Concerns:**

None.

**Quality:**

2

**Strengths And Weaknesses:**

Motivated by the observation that the activations of some neurons are strongly correlated with task performance, the paper introduces the notion of RL skill neurons. To retain the performance of skill neurons, their updates are limited with a gradient masking technique.


S1: The paper tackles an interesting challenge.
S2: The paper is well-written in general.
S3: Experimental results show improvements over baselines.
S4: The code is publicly available.

W1: Experiments are conducted on 2 benchmarks only. Moreover, these benchmarks are rather simple (Atari, Meta-World)
W2: The discussion of the most related works is very short.
W3: The method is somewhat ad hoc and the innovativeness is limited.
W4: The superparameter alpha is just set to 0.2 without explaination (Line 204)

Ad W1: There are a lot of more complex benchmarks for RL, e.g., MuJoCo and DeepMind. It is not clear if the method works there.
Ad W2: There are some RL approaches, that also focus on the neuron level. These are only extremely briefly discussed in Lines 109 - 113. Experiments lack comparison to some of those most relevant methods.
Ad W3: Gradient masking and the identification of task-specific neurons are not novel ideas but already known from other works. The same holds for periodic replays during training. The specific combination for reinforcement learning is new but also quite straightforward and not well motivated. There are no theoretical guarantees when this method works and only limited experiments.
Ad W4: There is no explaination, theoretical foundation or guideline on how to set this parameter. It is only experimentally validated.

---

> ### Author Rebuttal · Authors · 2025-07-31
>
> We appreciate your valuable feedback on the benchmark, related work, innovativeness, hyper-parameter, and theoretical explaination of our paper. We carefully address your concerns in the following responses.
>
>   Q1: more complex benchmark
>
>   A1: Thanks for the suggestion. To further validate the generalization ability, we apply NBSP to the **DeepMind Control Suite (DMC)**, a widely recognized benchmark for continuous control tasks built on the MuJoCo physics engine. Our evaluation protocol for DMC mirrors that used on Atari, as detailed in lines 239–248 and 672–676 of the paper. We conduct cycling training on two groups of sequential tasks within DMC, reporting the standard metrics: Average Return (AR), Forgetting Measure (FM), and Forward Transfer (FWT).  The results in the following table show that NBSP greatly reduces FM compared to vanilla SAC, indicating higher stability. NBSP also achieves a higher AR and FWT, indicating better plasticity. These results further support that **NBSP performs robustly in more complex benchmark like DMC, and could balance stability and plasticity beyond benchmarks like Atari or Meta-World.**
> |cycling sequential tasks|metric|vanilla SAC|NBSP|
> |:--|:--|:--|:-|
> |(Cartpole Swingup → Cartpole Balance)|AR $\uparrow$|746.80±5.26|843.47±11.39|
> |(Cartpole Swingup → Cartpole Balance)|FM $\downarrow$|307.42±16.41|59.26±10.89|
> |(Cartpole Swingup → Cartpole Balance)|FWT $\uparrow$|874.63±8.22|883.32±6.69|
> |(Walker Walk → Walker Stand) |AR $\uparrow$|790.26±54.58|861.09±24.99|
> |(Walker Walk → Walker Stand) |FM $\downarrow$|272.26±67.62|170.63±38.12|
> |(Walker Walk → Walker Stand) |FWT $\uparrow$|899.44±26.46|914.59±30.07|
>
> Q2: the discussion of more related works
>
>   A2: Thank you for the suggestion.
>
>   (1) We have expanded the related work section as follows:
>
>   "Despite these achievements, the exploration of skill neurons in DRL remains largely under-explored. Some work focus on task-specific sub-network selection in the neuron-level, such as CoTASP and PackNet. At a finer granularity, other approaches target individual neuron management. NPC uses a normalized Taylor criterion to identify and constrain important neurons, primarily to maintain stability. Similarly, ReDO [1] and its successor GraMa [2] introduce schemes for dormant neuron management and activity quantification based on activation and gradient magnitudes, aiming to improve network expressive power and recoverability. NE [3] further propose to dynamically adapt network topology through neuron growth and pruning based on potential gradients. However, these methods overloop the fundamental link between a neuron and the task's goal in reinforcement learning, and identify neurons through static measures like activation or gradient. Our work addresses this critical gap and moves beyond the static measures by discovering and leveraging a significant correlation between a neuron's activation and the specific objective of the reinforcement learning task. This insight allows us to identify the underlying critical neurons, those that are functionally relevant to the task's success."
>
>  (2) We include comparisons to these neuron-level approaches in our experiments:
> - We have compared NBSP with CoTASP and NPC (Table 1), evaluate our neuron identification method against activation- and gradient-based alternatives used in ReDO and GraMa (Table 3), and note that PackNet is not directly comparable due to its reliance on task identity at inference, which contradicts our task-agnostic setting.
>
>  - We add a new comparison experiment between NBSP and NE using the cyclic task sequence (button-press-topdown → window-open). The results are presented below. NE maintains high plasticity (reflected by its FWT score) but suffers from significant stability loss (high FM). And NBSP outperforms NE on all the metrics.
>     |method| ASR $\uparrow$|FM $\downarrow$|FWT $\uparrow$|
>     | :-- | :--| :--| :-- |
>     |NE| 0.60 ± 0.03 | 1.00 ± 0.00 | 0.94 ± 0.05 |
>     |NBSP| 0.95 ± 0.05 | 0.08 ± 0.12 | 0.98 ± 0.01 |
>
>   Q3:  the innovativeness of the method
>
>   A3:  Thanks for the question. We would like to highlight that **the primary contribution of our work is the identification of RL skill neurons and the demonstration of their crucial role in balancing stability and plasticity in reinforcement learning**, which opens opportunities for addressing the stability-plasticity dilemma from the perspective of neurons.
>
>   The integration of various components in our method is not a simple combination of techniques, but a cohesive framework built on a clear and fundamental principle: achieving a balance between stability and plasticity by understanding and controlling the behavior of individual neurons. **The entire methodology follows a logical and step-by-step progression**:
>
>   1. **Foundational motivation**:
>
>      The stability and plasticity of the agent are closely related to expressive capabilities of network, which are fundamentally influenced by the behavior of its individual neurons. We are inspired by prior work showing that specific neurons can encode specialized information, leading us to hypothesize the existence of skill neurons in DRL, which are crucial for task-relevant information. Our core idea is to leverage these specific neurons as the key to balancing stability and plasticity at the most fundamental neuron level.
>   2. **Validation and identification**:
>
>      The straightforward way to validate the relevance of a neuron to a task is to observe how its activity directly contributes to the success. This insight guides us to observe how neuron activation contributes to the success of a task, which reveals the existence of RL skill neurons. Built upon this key observation, we propose the novel neuron identification method.
>   3. **Adaptive gradient masking**:
>
>      To maintain stability, it's essential to preserve the knowledge of previous tasks, which is encoded by the RL skill neurons. Thus we propose gradient masking to restrict changes in their activation patterns. However, simply freezing RL skill neurons would hinder the agent's ability to adapt to new tasks. To address this issue, we propose the adaptive gradient masking technique according to the scores of the RL skill neurons to enable the fine-tuning of these neurons.
>   4. **Experience replay**：
>
>      With the RL skill neurons partially constrained, the remaining neurons are primarily responsible for learning new tasks and ensuring plasticity. However, this can lead to large changes in their activation patterns, which may severely impair performance on previously learned tasks. To mitigate the risk of the agent becoming overly biased towards a new task, which could compromise overall stability, we integrate experience replay.
>
>   Additionally, we advance current techniques in that (1) For gradient masking, we adopt a **soft, adaptive masking mechanism**, where each RL skill neuron is assigned a dynamically adjusted mask value based on its score. This ensures that critical information encoded by RL skill neurons is protected from interference during new task learning while still allowing a degree of fine-tuning for better adaptation.  (2) For neuron identification, we **introduce Goal Metric (GM) as an evaluation criterion and score each neurons according to the correlation between their relative activation and the relative GM**, and then select the RL skill neurons with higher scores. This approach enables a more accurate evaluation of neurons, helping to identify those that truly encode critical task-related skills.
>
>   Q4: the hyper-parameter $\alpha$
>
>   A4:  Thank you for the question. **$\alpha$ controls the strength of the adaptive gradient masking by scaling the mask value applied to RL skill neurons.** A smaller $\alpha$ imposes stronger restrictions on these neurons, enhancing stability but potentially impairing plasticity. When $\alpha = 0$, the method degenerates into hard gradient masking and loses its adaptive property.
>
>   We conduct ablation experiments varying its value from 1.0 to 0.1 using the cyclic task sequence (button-press-topdown → window-open) . As $\alpha$ decreases, FM consistently improves (lower is better), confirming enhanced stability. Meanwhile, FWT remains relatively stable, suggesting plasticity is not highly sensitive to $\alpha$. Notably, we observe strong ASR performance as long as $\alpha < 0.3$, indicating **NBSP is robust to the choice of $\alpha$ in a reasonable range**.
>   | $\alpha$ | ASR $\uparrow$|FM $\downarrow$|FWT $\uparrow$|
>   | :- | :- | :- | :-|
>   | 0.1| 0.93 ± 0.04 | 0.07 ± 0.10 | 0.96 ± 0.01 |
>   |0.2| 0.95 ± 0.05 | 0.08 ± 0.12 | 0.98 ± 0.01 |
>   |0.3| 0.91 ± 0.07 | 0.13 ± 0.16 | 0.98 ± 0.01 |
>   |0.5| 0.85 ± 0.01 | 0.33 ± 0.00 | 0.98 ± 0.01 |
>   |1.0| 0.81 ± 0.06 | 0.48 ± 0.15 | 0.98 ± 0.01 |
>
>   Q5: theoretical explainations
>
>   A5: Thanks for the insightful question. We acknowledge that there is no formal theoretical explanation for NBSP currently, and we believe this presents an important future direction for this research area. Empirically, **recent research has uncovered the diverse functional roles of individual neurons in deep neural networks**, including their ability to store factual knowledge, exhibit language-specific activation patterns, and capture safety-related features. Inspired by these findings, we hypothesize that task-specific skill neurons exist in RL and can be leveraged to balance stability and plasticity. **Our extensive experimental results provide strong evidence and empirical value for the effectiveness of NBSP in achieving this balance**.
>
>   [1] Sokar G, et al. "The dormant neuron phenomenon in deep reinforcement learning."(2023).
>
>   [2] Liu J, et al. "Measure gradients, not activations! Enhancing neuronal activity in deep reinforcement learning." (2025).
>
>   [3] Liu J, et al. "Neuroplastic expansion in deep reinforcement learning." (2024).

---

### Official Review · Reviewer_gsYq · 2025-07-03

**Clarity:** 4
**Significance:** 3
**Originality:** 3
**Rating:** 5
**Confidence:** 4

**Summary:**

This paper explores how individual neuron in the artificial neural networks can improve the stability and plasticity trade-offs in the continual reinforcement learning setting. The criteria on which neurons are relevant for specific task is based on the comprehensive score defined by the authors, which take into account of the activations of the neuron. In order to preserve the activity of the neuron across the different task to limit interference and forgetting, the model uses an approach known as adaptive gradient masking, which mask the gradients of the skill neurons in order to restrict changes in their activity.

**Questions:**

1. According to Figure 1, it seems that higher activation contributes more to success of the task. However, can the inverse also be true, such that high activations lead to unsuccessful outcome of the task? How robust or consistent is this effect?

2. How critical is it to use the experience replay buffer with information encountered from previous tasks?

3. Line 163: What do the authors mean by average step? Is this corresponding to the number of steps per task?

4. Does the experience replay store information on previously encountered tasks?

**Ethical Concerns:**

["NO or VERY MINOR ethics concerns only"]

**Final Justification:**

I maintained my score because after the rebuttal, the limitation of this work is that the algorithm still relies (heavily) on the use of replay buffer to mitigate forgetting. But since the authors agree to clarify this point in the revised manuscript, I did not feel the need to downgrade the rating.

**Limitations:**

Yes

**Quality:**

3

**Strengths And Weaknesses:**

# Strength
1. Overall, the paper is well-written and easy to read. The authors focused on a less studied area of the
2. Figure 1 was helpful to see how the magnitude of the activation correlates to the successful or failure outcome of the task.
3. The results in the Metaworld and Atari benchmark looks promising. It showed that the proposed algorithm was able to outperform several competitive baseline models such as Continual backprop and plasticity injection.
4. The proposed model does not require resetting parameters in the neural network. This approach is similar to EWC, which focuses on the synaptic connection (weights) between the neurons, rather than the activity of the neuron itself.
5. Ablation studies have been performed the importance of the use of both gradient masking and experience replay

# Weakness
1. Requires the sweeping the hyper-parameter to determine the proportion of the skill neurons
2. Seems to require the use of an experience replay buffer that contains information from the previous tasks

---

> ### Author Rebuttal · Authors · 2025-07-31
>
> We appreciate your valuable feedback on the hyper-parameter, experience replay technique, and your acknowledgement of novelty, writing, and experimental validation of our paper. We will address each of your comments and concerns in the following responses.
>
>   Q1: hyper-parameter $m$
>
>   A1: Thanks for your insightful comment regarding the impact of the RL skill neuron proportion ($m$) on NBSP's performance. As illustrated in Fig.4, we observe that NBSP's performance generally improves with increasing $m$, up to a certain threshold, after which it tends to decline. This behavior highlights the role of $m$ in balancing stability and plasticity. Crucially, our experiments demonstrate that **NBSP consistently delivers ASR improvements across a reasonably broad range of $m$ values (from 0.15 to 0.3)**. This indicates that **the selection of $m$ is not overly sensitive. And a $m$ within a practical and operational range can achieve strong performance gains across different tasks and benchmarks, as shown in our current experiments**. We believe that fine-grained tuning within this robust range would be straightforward for practical applications. And we acknowledge that adaptively determining $m$ could further enhance practical convenience and optimize performance, and presents a promising direction for future work, which has also been discussed in the appendix.
>
>   Q2: require the use of an experience replay buffer
>
>   A2: Thanks for your suggestion. Our findings in Table 4 show that **experience replay is indispensable for ensuring the stability**. While **gradient masking alone could alleviate stability loss** to a certain degree compared to vanilla SAC, its performance still falls behind the combined strategy using experience replay. We would like to highlight that these two mechanisms actually complement each other. (1) **Gradient masking primarily targets the RL skill neurons**, helping to reduce interference with past knowledge while preserving the ability to fine-tune for new tasks. (2) On the other hand, **experience replay mainly acts on neurons except the RL skill neurons**, preventing them from becoming overly biased toward recent tasks. Thus, they work in a synergistic way to contribute to the balance between stability and plasticity.
>
>   Q3: activation and the success of the task
>
>   A3: Thanks for this insightful question. **There do exist RL skill neurons that show higher activation during failed episodes**. We summarize the proportion of these two types of neurons in the following table.
>
>   |         task         | higher activation -> success | lower activation -> success |
>   | :------------------ | :--------------------------: | :-------------------------: |
>   | button-press-topdown |            70.4%             |            29.6%            |
>   |     window-open      |            72.2%             |            27.8%            |
>   |     window-close     |            71.5%             |            28.5%            |
>
>   We further look into the average activation of these two groups of neurons across three different tasks.
>
>   - **Neurons correlated with higher activation for success exhibit much higher average activation during successful trials compared to failed ones.**
>
>     |         task         | success | failure |
>     | :------------------ | :-----: | :-----: |
>     | button-press-topdown |  0.344  |  0.148  |
>     |     window-open      |  0.384  |  0.164  |
>     |     window-close     |  0.356  |  0.140  |
>
>   - In contrast, **neurons correlated with lower activation for success show higher average activation during failed trials.**
>
>     |         task         | success | failure |
>     | :------------------ | :-----: | :-----: |
>     | button-press-topdown |  0.032  |  0.240  |
>     |     window-open      |  0.040  |  0.261  |
>     |     window-close     |  0.044  |  0.216  |
>
>   Overall, the RL skill neurons we identify exhibit a clear distinction in activation patterns between successful and failed episodes, as reflected in the summary statistics. This highlights their critical role in task performance.
>
>
>   Q4: average step
>
>   A4: Thanks for the clarification. **The "average step" refers to a fixed size (set to 5e4 in our experiments) over which we compute the average activation and Goal Metric (GM),** which is used as a baseline for calculating the over-activation rate. It is a hyper-parameter and **does not correspond to the number of steps per task**.
>
>   Q5: does the experience replay store information on previously encountered tasks?
>
>   A5: Thanks for this insightful question, the experience replay buffer stores information from all previously encountered tasks. We investigate this problem by experimenting on the cycling task sequence (button-press-topdown → window-close → door-open → drawer-close), where we compared two buffer configurations:
>
>   (1) Storing only the most recent task's experience.
>
>   (2) Storing experience from all past tasks.
>
>   As the table below clearly shows, **restricting the buffer to storing only the most recent task leads to significantly higher FM values, indicating greater forgetting and reduced stability. In comparison, storing data from all previously encountered tasks consistently improves stability**.
>
>   |             buffer configuration              |     ASR $\uparrow$     |     FM $\downarrow$     |     FWT $\uparrow$    |
>   | :-------------------------------------------- | :---------: | :---------: | :---------: |
>   | storing only the most recent task's experience | 0.51± 0.12  | 0.75 ± 0.12 | 0.90 ± 0.20 |
>   |    storing experience from all past tasks.     | 0.74 ± 0.07 | 0.34 ± 0.15 | 0.95 ± 0.06 |
>
>   To further confirm this, we evaluate the success rates of the first three tasks after training on the fourth task, specifically under the setting where the experience replay buffer contains only data from the most recent task. Results in the following table show that when the agent is trained on the four task with the buffer restricted to experience from only the third task, it **retains its ability to perform the third task but entirely fails on the first and second one.** This validates that: **the experience replay buffer must store information from all previously encountered tasks.** Omitting earlier tasks from the buffer directly leads to catastrophic forgetting.
>
>   |     task     | first task: button-press-topdown | second task: window-close | third task: door-open |
>   | :----------: | :------------------------------: | :-----------------------: | :-----------------------: |
>   | success rate |           0.00 ± 0.00            |        0.03 ± 0.05        |  0.80 ± 0.08        |

---

> ### Author Response · Authors · 2025-08-02
>
> We would like to thank the reviewer for this valuable insight, which further clarifies the role of RL skill neurons. And we will make modifications to the revised manuscript accordingly to better clarify this point.
>
> (1) We agree that RL skill neurons cannot resolve catastrophic forgetting on their own. For example, for the success of a specific task, those non-RL skill neurons could also matter, which may relate to factors such as the environments and contexts. And catastrophic forgetting of these non-RL skill cabilities could also lead to task failures. We believe neuron-level research for generalization across tasks in a more general sense would be an important future direction. For example, we may build the agent upon a well-pretrained foundation model that already masters adequate world knowledge, and translate the balance of stability and plasticity into maintaining existing world knowledge and learning new skills, where RL skill neurons could still help by mitigating catastropic interference and more nuanced neuron-level control might be introduced by taking into account those world knowldge.
>
> (2) We agree that under the current framework, the primary contribution of RL skill neurons is mitigating catastrophic interference by preventing overwriting the knowledge encoded in these neurons when learning new tasks. We would like to highlight that mitigating catastrophic interference could also benefit catastrophic forgetting, where RL skill neurons and experience replay buffer work in a synergic way to alleviate forgetting. As shown in Table 2, the agent with only experience replay still suffers from catastrophic forgetting, while by incorporating NBSP with experience replay, we improve catastrophic forgetting by a large margin (FM 0.50 ± 0.16 -> 0.08 ± 0.12).

---

> > ### Comment · Reviewer_gsYq · 2025-08-04
> >
> > Thank you for the clarification. Please include the information in (1) and (2) in the manuscript, particular on (2). I do not have any further questions and have also read through reviews from other reviewers. I have decided to maintain my score as it is.

---

> > > ### Author Response · Authors · 2025-08-05
> > >
> > > Thanks a lot for your efforts in reviewing this paper and proposing these valuable insights and suggestions. We will include discussions of both (1) and (2) of the above discussion into our revised manuscript.

---

### Note · Authors · 2025-08-13

First of all, we sincerely thank all reviewers for their diligent reviews and constructive feedbacks. We are encouraged that the reviewers acknowledged the novelty, writing quality, and empirical validation of our work.

Following the reviewers' suggestions, we have primarily conducted the following additional experiments:

- As suggested by Reviewer gsYq:
  - We investigated if experience replay retains information from previous tasks.
- As suggested by Reviewer yS8E:
  - We applied NBSP to the more complex DMC benchmark to further validate the generalization ability.
- As suggested by Reviewer 4Ero:
  - We extended the compared baselines to include Shrink & Perturb, NE, and UPGD, all of which aim to balance stability and plasticity.
  - We incorporated experience replay into ANCL and CoTASP for a fairer comparison.
  - We implemented NBSP on the on-policy algorithm PPO in addition to SAC.
- As suggested by Reviewer AmTw:
  - We further investigated the effect of task order.
- As suggested by Reviewer gsYq, yS8E, 4Ero and AmTw:
  - We presented the sensitivity analysis of NBSP's hyperparameters.


In addition to these supplementary experiments, we have also provided detailed explanations to address the reviewers' concerns:

- As suggested by Reviewer gsYq:
  - We clarified the relationship between neuron activation and task success.
  - We clarified the role of RL skill neurons in catastrophic interference.
  - We explained the necessity of using experience replay.
- As suggested by Reviewer yS8E:
  - We clarified the innovativeness of the method.
- As suggested by Reviewer 4Ero:
  - We included a discussion on the entropy bonus of PPO.
- As suggested by Reviewer AmTw:
  - We provided a discussion on the relationship between NBSP and other neuron-level plastic neural network models.
- As suggested by Reviewer yS8E and 4Ero:
  - We complemented the discussion of the RL approaches that focus on the neuron level and the balance between stability and plasticity.

We hope these experiments and clarifications could further strengthen our paper and satisfactorily address the raised concerns. And we will incorporate them into the revised manuscript. Thank you again for your valuable time and expertise.

Lastly, we would also like to thank all ACs for their significant efforts and contributions.

Best Regards,

Authors of Paper 10315

---

### Decision · Program_Chairs · 2025-09-17

**Decision:**

Reject

**Comment:**

This paper identifies task-specific skill neurons in a neural network which, if frozen after a task change in a multi-task sequential learning protocol, preserve performance on the old task without significantly slowing down training on a new task. While similar observations regarding the localization of learning to specific neurons have been made in prior works, this paper is the first to show that a relatively straightforward neuron freezing protocol can effectively preserve knowledge in the multi-task reinforcement learning domain. The fact that learning can be localized in this way in RL, and that the number of neurons needed to preserve prior task performance is often quite low, is a surprising and valuable contribution to the literature, and reviewers agreed during the discussion that this was the paper's primary source of merit.

While the paper makes an interesting empirical observation, I agree with reviewers that there were significant weaknesses in the positioning of the method and the fairness of the empirical evaluations, and do not believe that the author response in fact sufficiently addressed these issues. The evaluation protocols are non-standard in that they consider only a small number of task changes, which is precisely the setting where a method like this would be expected to shine but do not reflect many naturally-arising learning problems, and largely compare against methods which target only one of stability or plasticity. It is acceptable to conduct this type of evaluation in order to provide an “existence proof” of the method achieving its goal. However, it is also important to acknowledge the limitations of the proposed method, and it would improve the soundness of the paper’s scientific contribution to also include an evaluation on a standard continual RL benchmark such as continual world (sequential metaworld). continual atari, or at least some variant of the existing evaluations that considers an arbitrarily long sequence of tasks – even if the NBSP method does not outperform the baselines that were designed to target these benchmarks.

I would also encourage the authors to expand their discussion of related work with regards to both existing methods for localizing learning in neural networks and maintaining plasticity. For example, [1] discusses localization of learning to particular neurons in a network and provides a nice motivation for the idea of targeting specific neurons in RL. While the paper mentions some resetting-based approaches, many of the most effective methods for maintaining plasticity leverage normalization [2, 3] and regularization [4] (while Kumar et al. is cited, their approach was not designed for RL and requires careful tuning to work in reinforcement learning settings for reasons outlined in [5]), and it would be good to at least discuss why these approaches would be unlikely to help in the problem setting studied in this paper (for example, I could imagine that layer normalization might reduce the efficacy of the gradient masking due to the increased correlation between neurons, which [6] notes as reducing the efficacy of ReDO).

[1] Maini, Pratyush, et al. "Can neural network memorization be localized?." arXiv preprint arXiv:2307.09542 (2023).

[2] Ball, Philip J., et al. "Efficient online reinforcement learning with offline data." International Conference on Machine Learning. PMLR, 2023.

[3] Lyle, Clare, et al. "Understanding plasticity in neural networks." International Conference on Machine Learning. PMLR, 2023.

[4] Lewandowski, Alex, et al. "Learning continually by spectral regularization." arXiv preprint arXiv:2406.06811 (2024).

[5] Lyle, Clare, et al. "Normalization and effective learning rates in reinforcement learning." Advances in Neural Information Processing Systems 37 (2024): 106440-106473.

[6] Liu, Jiashun, et al. "Measure gradients, not activations! Enhancing neuronal activity in deep reinforcement learning." arXiv preprint arXiv:2505.24061 (2025).